# A moonlighting function of a chitin polysaccharide monooxygenase, CWR-1, in *Neurospora crassa* allorecognition

Tyler C Detomasi[1†], Adriana M Rico-Ramírez[2†], Richard I Sayler[3], A Pedro Gonçalves[2‡], Michael A Marletta[1,3,4], N Louise Glass[2]*

[1]Department of Chemistry, University of California, Berkeley, Berkeley, United States; [2]Department of Plant and Microbial Biology, University of California, Berkeley, Berkeley, United States; [3]California Institute for Quantitative Biosciences, University of California, Berkeley, Berkeley, United States; [4]Department of Molecular and Cell Biology, University of California, Berkeley, Berkeley, United States

**\*For correspondence:**
Lglass@berkeley.edu

[†]These authors contributed equally to this work

**Present address:** [‡]Department of Cell Biology and Anatomy, College of Medicine, National Cheng Kung University, Tainan City, Taiwan

**Competing interest:** The authors declare that no competing interests exist.

**Abstract** Organisms require the ability to differentiate themselves from organisms of different or even the same species. Allorecognition processes in filamentous fungi are essential to ensure identity of an interconnected syncytial colony to protect it from exploitation and disease. *Neurospora crassa* has three cell fusion checkpoints controlling formation of an interconnected mycelial network. The locus that controls the second checkpoint, which allows for cell wall dissolution and subsequent fusion between cells/hyphae, *cwr* (cell wall remodeling), encodes two linked genes, *cwr-1* and *cwr-2*. Previously, it was shown that *cwr-1* and *cwr-2* show severe linkage disequilibrium with six different haplogroups present in *N. crassa* populations. Isolates from an identical *cwr* haplogroup show robust fusion, while somatic cell fusion between isolates of different haplogroups is significantly blocked in cell wall dissolution. The *cwr-1* gene encodes a putative polysaccharide monooxygenase (PMO). Herein we confirm that CWR-1 is a C1-oxidizing chitin PMO. We show that the catalytic (PMO) domain of CWR-1 was sufficient for checkpoint function and cell fusion blockage; however, through analysis of active-site, histidine-brace mutants, the catalytic activity of CWR-1 was ruled out as a major factor for allorecognition. Swapping a portion of the PMO domain (V86 to T130) did not switch *cwr* haplogroup specificity, but rather cells containing this chimera exhibited a novel haplogroup specificity. Allorecognition to mediate cell fusion blockage is likely occurring through a protein–protein interaction between CWR-1 with CWR-2. These data highlight a moonlighting role in allorecognition of the CWR-1 PMO domain.

## Editor's evaluation

This fundamental study identifies an important role for a lytic polysaccharide monooxygenase in allorecognition in the filamentous fungus *Neurospora crassa*, which is independent of the catalytic activity of this remarkable class of proteins. The study's findings are compelling, combining microscopy with genetics and biochemistry. The study will be of great interest to fungal biologists and microbiologists, as well as biochemists studying carbohydrate-active enzymes.

## Introduction

Allorecognition is the ability of a cell to recognize self or kin and has widespread importance for many organisms that have multicellular organization. Allorecognition in some form is utilized by the social amoeba, *Dictyostelium discoideum* (*Kundert and Shaulsky, 2019*; *Kuzdzal-Fick et al., 2011*),

the bacterium, *Proteus mirabilis* (***Gibbs et al., 2008***; ***Gibbs and Greenberg, 2011***), gymnosperms (***Pandey, 1960***), slime molds (***Clark, 2003***; ***Shaulsky and Kessin, 2007***), and invertebrates, such as *Botryllus schlosseri* (***Detomaso et al., 2005***; ***Rosengarten and Nicotra, 2011***; ***Yoshito et al., 2008***) or the cnidarian *Hydractinia symbiolongicarpus* (***Rosengarten and Nicotra, 2011***). In *Drosophila melanogaster*, the *dscam* gene, which contains >19,000 splicing isoforms, uses differential splicing and homo-dimer binding to recognize and not synapse with itself (***Wojtowicz et al., 2004***; ***Wojtowicz et al., 2007***). In vertebrate species, the major histocompatibility complex (MHC) is crucial for the immune response and allows cells to identify infected cells or cells that are no longer kin for destruction (***Afzali et al., 2008***; ***Marino et al., 2016***).

Although not strictly multicellular, most filamentous fungi grow as an interconnected hyphal network that shares cytoplasm, nuclei, and nutrients, and utilizes allorecognition machinery to regulate this process (***Gonçalves and Glass, 2020***). This syncytial organization of filamentous fungi allows an interconnected colony to thrive in an environment that is heterogeneous for nutrients and habitats (***Anna et al., 2012***). The formation of an interconnected mycelial network is a process that is controlled by communication and cell fusion checkpoints to help ensure that cells with higher genetic identity form a colony (***Gonçalves et al., 2020***; ***Gonçalves et al., 2020***). Without checkpoints, cell fusion can occur between genetically unrelated colonies and cells, allowing 'cheater' nuclei to steal resources (***Bastiaans et al., 2016***; ***Grum-Grzhimaylo et al., 2021***) or enabling the spread of cytoplasmic diseases such as mycoviruses (***Zhang and Nuss, 2016***) or selfish genetic elements (***Debets et al., 2012***), which are spread via cell fusion throughout fungal populations.

The filamentous ascomycete species, *Neurospora crassa*, has three cell fusion checkpoints that function in allorecognition. The first checkpoint regulates chemotropic interactions between hyphae and/or germinated asexual spores (germlings) and is controlled by the determinant of communication (*doc*) locus that senses chemosignaling (***Heller et al., 2016***). Cells with identical allelic specificity at the *doc* locus undergo chemotropic interactions, while cells with alternate *doc* allelic specificity show greatly reduced interactions. The second checkpoint is regulated by the cell wall remodeling locus, *cwr*, which contains the two linked genes, *cwr-1* (UniProtKB Q1K703) and *cwr-2* (UniProtKB Q1K701) (***Gonçalves et al., 2019***). Cells/hyphae with identical allelic specificity at *doc* and *cwr* undergo chemotropic interactions and, upon contact, undergo cell wall dissolution, membrane merger, and cytoplasmic mixing. However, if cells have identical *doc* specificity, but differ in *cwr* allelic specificity, hyphae/cells undergo chemotropic interactions, but upon contact, cells remain adhered and fail to undergo cell wall deconstruction and membrane merger at the point of contact (***Gonçalves et al., 2019***). The final checkpoint is a post-fusion checkpoint. Following chemotrophic interactions (*doc* alleles with identical allelic specificity) and cell fusion (*cwr* alleles with identical allelic specificity), if cells/hyphae differ in specificity at several post-fusion allorecognition loci, the fusion cell is compartmentalized and undergoes rapid cell death (***Daskalov et al., 2019***; ***Heller et al., 2018***; ***Rico-Ramírez et al., 2022***). Recent data in *N. crassa* showed that one of these post-fusion allorecognition loci has functional and structural similarity to mammalian gasdermin and confers rapid cell death via a pyroptotic-like mechanism (***Corinne et al., 2022***; ***Daskalov and Glass, 2022***; ***Daskalov et al., 2020***; ***Rico-Ramírez et al., 2022***) while in the related filamentous fungus, *Podospora anserina*, a different post-fusion cell death allorecognition locus has functional similarities to necroptosis (***Saupe, 2011***; ***Saupe, 2020***).

The *cwr-1* gene is predicted to encode a chitin-active polysaccharide monooxygenase (PMO) in the CAZy (Carbohydrate-Active enZYme) database designation of Auxiliary Activity 11 (AA11) family (***Levasseur et al., 2013***). The *cwr-2* gene encodes a predicted protein that contains eight transmembrane regions with two annotated domains of unknown function (PF11915). Within populations of *N. crassa*, *cwr-1/cwr-2* alleles show severe linkage disequilibrium and fall into six different haplogroups (HGs). Cells/germlings containing only CWR-1 from HG1 are blocked in cell fusion with isogenic cells containing only CWR-2 from a different haplogroup (***Gonçalves et al., 2019***). These data indicate that incompatible CWR-1-CWR-2 function in *trans* when present in different interacting cells and are necessary and sufficient to trigger the cell fusion block.

PMOs, alternatively referred to as lytic polysaccharide monooxygenases (LPMOs), have been studied extensively for over a decade (***Phillips et al., 2011***; ***Vaaje-Kolstad et al., 2010***). These proteins were first discovered as auxiliary redox enzymes that greatly enhance cellulose degradation in combination with glycosyl hydrolases (***Harris et al., 2010***). Beyond just cellulolytic activity, other

PMOs have been demonstrated to show activity on other polysaccharides such as chitin (*Hemsworth et al., 2014*), starch (*Vu et al., 2014b*; *Lo Leggio et al., 2015*; *Vu et al., 2019*; *Vu and Marletta, 2016*), xylans (*Couturier et al., 2018*; *Hüttner et al., 2019*), and various hemicelluloses (*Agger et al., 2014*; *Monclaro et al., 2020*). PMOs have been of fundamental interest because they catalyze the hydroxylation of a strong C–H bond and their utility in industrial biofuel and nanocellulose applications (*Johansen, 2016*; *Moreau et al., 2019*). However, as more PMO families have been identified, other roles have been noted (*Vandhana et al., 2022*; *Hangasky et al., 2020*), including in the life cycle of insect viruses (*Chiu et al., 2015*), insect development (*Sabbadin et al., 2018*), fungal development (*Fu et al., 2014*; *Gonçalves et al., 2019*; *Maddi et al., 2012*), and in symbiotic associations between bacteria and arthropods (*Distel et al., 2011*; *Pinheiro et al., 2015*). A family of related proteins termed X325 have a similar fold and bind copper but are not functional PMOs. Instead one member of this family has a unique biological role in copper transport (*Garcia-Santamarina et al., 2020*).

All PMOs contain a secretion signal peptide that is cleaved, leaving the mature protein with an N-terminal histidine residue that binds a single copper atom (*Phillips et al., 2011*). Of the four other proteins characterized in the AA11 family, only a chitin-degradation role has been demonstrated (*Hemsworth et al., 2014*; *Rieder et al., 2021*; *Støpamo et al., 2021*; *Wang et al., 2018a*; *Wang et al., 2018b*). Three of the four proteins have a similar protein architecture composed of an N-terminal PMO domain followed by a glycine serine (GS)-rich disordered linker and an X278 domain at the C-terminus. The X278 domain has been compared to CBM sequences (*Hemsworth et al., 2014*). The recently published Alphafold structure in UniProt (UniProtKB Q1K703) shows two aromatic residues on the same side of a small predominant beta-sheet domain that may be positioned to bind polysaccharides. If this is a true CBM, it likely binds chitin as it is also found in GH18 chitinases (*Hemsworth et al., 2014*).

In this study, we assessed whether CWR-1 from each of the six different haplogroups has PMO activity and defined the substrate and products of that PMO activity. We also assessed whether CWR-1 proteins from each haplogroup formed unique products that would explain CWR-1 allelic specificity. By the further construction of PMO domain chimeras, we evaluated whether a polymorphic region of the PMO domain (LS and LC loop) was a factor in triggering allorecognition in interactions with CWR-2. Our data indicates that CWR-1 has chitin PMO activity, but that catalytic activity was not required for cell fusion blockage, highlighting an evolved and additional function of this PMO in allorecognition.

## Results

### The PMO domain of CWR-1 is sufficient to confer an allorecognition checkpoint

A multiple sequence alignment (MSA) of characterized AA11s revealed that three of the four characterized PMOs maintain a three-part architecture, while the other is a single-domain PMO containing only a catalytic domain (*Figure 1—figure supplement 1*). The three-domain architecture includes an N-terminus AA11 PMO domain *Hemsworth et al., 2014*; *Levasseur et al., 2013* followed by an extended GS-rich linker region that is likely disordered, and a C-terminal X278 domain proposed to be a chitin-binding domain; this domain is present in other predicted AA11 proteins and GH18 chitinases (*Gonçalves et al., 2019*; *Hemsworth et al., 2014*; *Figure 1A*). AA11 proteins, similar to all other PMOs, contain a signal peptide at the N-terminus (predicted by SignalP) that directs the protein to the ER and that is subsequently cleaved during translocation, leaving a histidine residue at the N-terminus.

CWR-1 is a member of the three-region architecture cluster, and many fungi contain both AA11 PMO architectures in their genome. To more fully examine the relationships of predicted AA11 proteins, we performed sequence similarity network (SSN) analysis using the EFI-EST BLAST feature with the CWR-1[HG1] sequence as the search peptide, which yielded ~1700 sequences. The three characterized AA11 proteins and CWR-1 that contained the GS linker and X278 domain were found in a single cluster (blue) (*Figure 1—figure supplement 2*). The single AA11 domain enzyme (*Af*AA11A) clustered separately into the red cluster. Since these domain architectures are clustered separately, this suggests a different biological function in fungi (*Figure 1—figure supplement 2*). This is not solely due to the difference in domain organization, for example, starch PMOs with different architectures

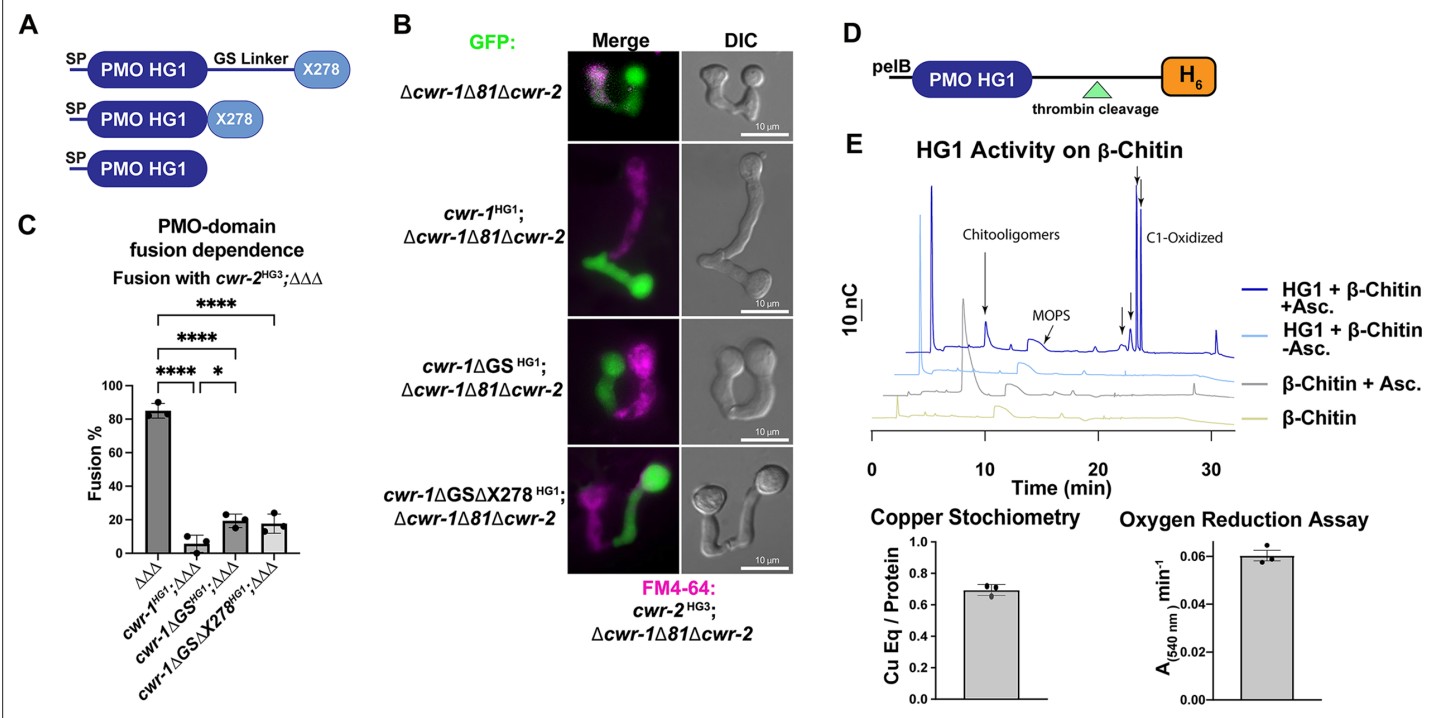

**Figure 1.** Characterization of the polysaccharide monooxygenase (PMO) domain from CWR-1 from a haplogroup 1 strain (FGSC2489) and functional dissection of CWR-1 domains *in vivo*. (**A**) A schematic diagram depicting the series of truncated *cwr-1* constructs studied. SP indicates signal peptide; GS linker indicates the glycine/serine-rich region that connects the PMO catalytic domain to the presumptive chitin-binding module, X278. (**B**) Cell fusion assays between *Δcwr-1Δ81Δcwr-2* germlings alone or bearing either HG1 *cwr-1* (FGSC2489) or truncated versions *cwr-1^ΔGS^, cwr-1^ΔGSΔX278^* (all expressing cytoplasmic GFP) and subsequently paired with FM4-64-stained *Δcwr-1Δ81Δcwr-2* germlings expressing an HG3 *cwr-2* allele (from JW258). (see *Figure 1—figure supplement 3* for fusion rates with all haplogroups [HGs] with HG1 and fusion controls). (**C**) Quantification of cell fusion frequencies shown in (**B**) of *Δcwr-1Δ81Δcwr-2* (GFP) germlings (*ΔΔΔ*), or *ΔΔΔ* germlings bearing HG1 *cwr-1^HG1^* or *ΔΔΔ* germlings bearing truncated versions *cwr-1^ΔG/S^* or *cwr-1^ΔGSΔX278^* and paired with FM4-64-stained *ΔΔΔ* germlings expressing an HG3 *cwr-2* allele. Experiments were performed in biological triplicate, assessing fusion of 100 germling pairs for each replicate. A one-way ANOVA followed by Tukey's post-hoc test was used for statistical analysis, error bars represent SD, *p<0.05, ****p<0.0001. Individual p-values are reported in *Figure 1—source data 4*. (**D**) Schematic depiction of the *E. coli* expression constructs. pelB indicates the signal peptide, the PMO domain from an HG1 strain (FGSC2489), and a thrombin-cleavable hexahistidine tag showing cleavage site at the indicated triangle (see *Figure 1—figure supplement 4* for protein gel and MS of purified protein). (**E**) Initial characterization of the PMO domain from HG1 strain (FGSC2489). This PMO exhibited C1-oxiding activity on chitin in the presence of ascorbate, reduced oxygen in the absence of chitin, and bound one copper atom per protein, all properties consistent with previously characterized AA11 PMOs. ICP analyses were done in technical triplicate, each datapoint in the oxygen reduction assay represents a biological replicate. All HPAEC-PAD assays were done in at least biological triplicate with a typical trace shown. Asc. means ascorbate. Black arrows denote peaks that elute in the region of the method corresponding to C1-oxidized oligosaccharides. The copper stoichiometry and oxygen reduction assay error bars are SEM where n=3. Source data for this figure can be found in *Figure 1—source data 1*, *Figure 1—source data 2*, and *Figure 1—source data 3*. See *Figure 1—figure supplement 5* for oxidized standards and *Figure 1—figure supplement 6* for MS/MS spectra on PMO products from α-chitin.

The online version of this article includes the following source data and figure supplement(s) for figure 1:

**Source data 1.** HPAEC-PAD source data.

**Source data 2.** ICP source data.

**Source data 3.** Horseradish peroxidase (HRP)-oxygen reduction assay source data.

**Source data 4.** p-Values.

**Figure supplement 1.** Alignment of characterized AA11s.

**Figure supplement 2.** AA11 sequence similarity network (SSN) annotated with characterized proteins.

**Figure supplement 2—source data 1.** Sequence similarity network (SSN) data from the EFI-EST tool as described in 'Materials and methods' used to construct the figure.

**Figure supplement 3.** Germling fusion phenotype and percentages among engineered stains.

**Figure supplement 3—source data 1.** p-Values.

**Figure supplement 4.** Purification and size of polysaccharide monooxygenase (PMO) domains of CWR-1 haplogroup (HG) proteins.

*Figure 1 continued on next page*

*Figure 1 continued*

**Figure supplement 4—source data 1.** Purified polysaccharide monooxygenase (PMO) domain from the six haplogroups (HGs).

**Figure supplement 4—source data 2.** Whole-protein MS data.

**Figure supplement 5.** Oligosaccharide standard chromatograms and elution times.

**Figure supplement 5—source data 1.** HPAEC-PAD source data.

**Figure supplement 5—source data 2.** HPAEC-PAD source data.

**Figure supplement 6.** MS/MS fragment is consistent with C1 oxidation.

**Figure supplement 6—source data 1.** Tandem MS source data.

cluster together at ~40% sequence ID (*Vu et al., 2019*). To determine which domain of CWR-1 was involved in allorecognition and the fusion checkpoint, a series of truncations were generated in the HG1 (FGSC2489) *cwr-1* allele: full-length *cwr-1*, removal of the GS linker, and removal of the GS linker and X278 domain (*Figure 1A*). Full-length and truncated constructs were inserted at the *his-3* locus under the native promoter (1111 bp) and transformed into a strain of *N. crassa* bearing a deletion of *cwr-1* (NCU01380) and *cwr-2* (NCU01382) (Δ*cwr-1Δ81Δcwr-2*) and expressing cytoplasmic GFP (*Gonçalves et al., 2019*). These strains were assessed for cell fusion in pairings with isogenic strains containing the *cwr-2* allele from each of the HG strains (HG1-6). The *cwr-1*$^{HG1}$ strain gave the strongest cell fusion block in pairings with *cwr-2*$^{HG3}$ germlings (*Figure 1—figure supplement 3A and B*). This *cwr-2*$^{HG3}$ strain was therefore used to test strains harboring the *cwr-1*$^{HG1}$ truncations that lacked the GS linker or the GS linker and the X278 domain. Strains containing any of the *cwr-1* truncations showed a block in cell fusion with the *cwr-2*$^{HG3}$ strain, including the truncation strain with only the catalytic PMO domain (*Figure 1B and C*). As a control, all of the truncation strains were also paired with a permissive mutant lacking *cwr-1* and *cwr-2* (Δ*cwr-1Δ81Δcwr-2*); all showed robust cell fusion frequencies (*Figure 1—figure supplement 3C and D*). These data showed that the PMO catalytic domain was sufficient to cause cell fusion arrest at the CWR allorecognition checkpoint.

Given that the PMO catalytic domain alone was responsible for conferring allorecognition, we performed experiments to probe the activity for this domain. The PMO catalytic domain of the *cwr-1*$^{HG1}$ allele was expressed in *Escherichia coli* using the periplasmic expression pelB system, which cleaves scarlessly during expression to ensure an N-terminal histidine with a C-terminal cleavable hexahistidine tag (*Figure 1D*) and subsequently purified using Ni-IMAC. The protein ran slightly higher (~26–27 kDa) on an SDS-PAGE gel than the predicted size of 22.6 kDa (*Figure 1—figure supplement 4A*); the mass was validated with whole-protein mass spectrometry. The spectrum revealed a deconvoluted mass of 22,624 Da, which corresponds to the exact mass of the protein with three disulfide bonds (22,630 − 3 × (2 Da for each disulfide)) (*Figure 1—figure supplement 4B*). The CWR-1$^{HG1}$ PMO domain bound ~0.7 equivalents of copper after reconstitution and was shown to reduce oxygen to hydrogen peroxide in the presence of ascorbate (*Figure 1E*). The activity of the CWR-1$^{HG1}$ protein was then tested with β-chitin as a substrate. New peaks between 18 and 22 min that corresponded to C1-oxidized products were only observed in the presence of both ascorbate and the CWR-1$^{HG1}$ PMO domain (*Figure 1E*). C1-oxidized standards were generated from chitooligosaccharides (*Figure 1—figure supplement 5A*) using the AA3, chitin-C1-oxidizing enzyme, ChitO (from *Fusarium graminearum*) (*Figure 1—figure supplement 5B*). The ChitO-C1-oxidized oligosaccharides eluted at timepoints similar to the new peaks from CWR-1$^{HG1}$ PMO at around 18–22 min (*Figure 1—figure supplement 5B*, *Figure 1E*). To further validate the CWR-1$^{HG1}$ C1 regioselectivity, tandem MS/MS was performed on the crude reaction from PMO$^{HG1}$ action on α-chitin in the presence of ascorbate. The resulting spectra were consistent with C1 oxidation (*Figure 1—figure supplement 6*). These data were in line with all other characterized AA11 PMOs that exhibit C1-oxidative activity on both α-chitin and β-chitin.

Our data showed that the PMO domain of CWR-1$^{HG1}$ was the driver for fusion arrest at the *cwr* allorecognition checkpoint. We therefore compared sequences of the PMO domain bioinformatically from the six different CWR-1 haplogroups. The phylogenetic tree showed identical clades to a tree created using full-length CWR-1 sequences (*Gonçalves et al., 2019*; *Figure 2A*). The intra-haplogroup similarity was very high with >95% sequence identity across the PMO domain for alleles within a haplogroup (*Supplementary file 1e*). All six PMO haplogroups contained the residues essential for PMO activity; two histidines in the histidine brace and residues implicated in the catalytic mechanism,

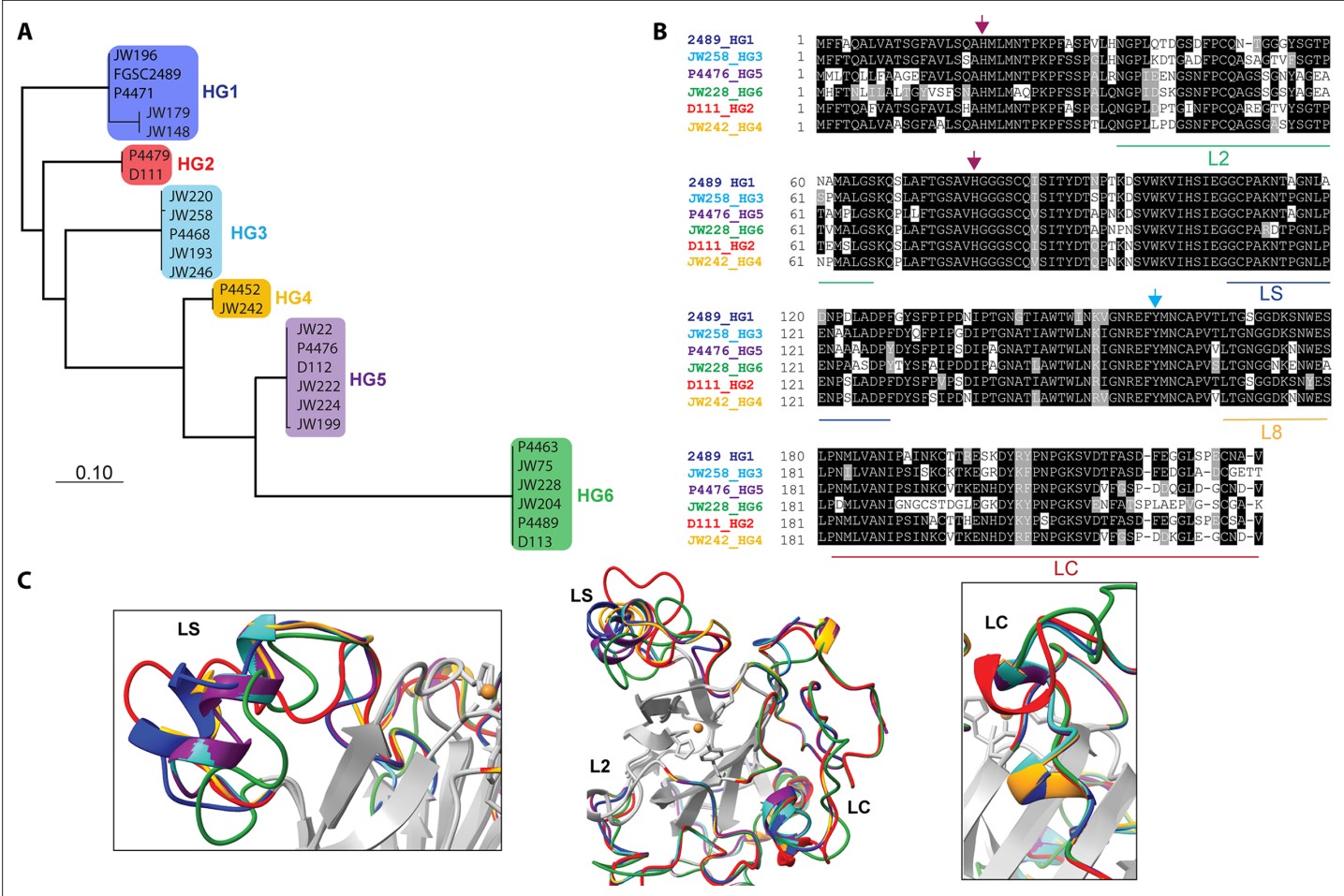

**Figure 2.** Comparison of CWR-1 polysaccharide monooxygenase (PMO) domains. In all panels, blue indicates *cwr-1* predicted proteins from HG1 isolates, red indicates HG2, light blue indicates HG3, orange indicates HG4, purple indicates HG5, and green indicates HG6 isolates. (**A**) A phylogenetic tree was constructed using the predicted CWR-1 PMO domain from 26 wild-type isolates. The phylogenetic tree was made using PhyML Phylogenetic Maximum Likelihood and edited in MEGA11. (**B**). Alignment of the six CWR-1 PMO domain protein sequences from representative isolates from HG1-6. The red arrows show the two histidine residues of the histidine brace that are involved in copper coordination. The blue arrow shows the position of the tyrosine involved in the secondary coordination sphere. L2, LS, and LC correspond to loops that exhibited sequence variation between the CWR-1 PMO domains among the six different haplogroups. (**C**) A SwissProt homology model of the PMO domain from the six haplogroups. There are apparent differences in the outer loops, whereas the core of the protein is not predicted to have significant differences between haplogroups. The four most affected loops are the loops that correspond to AA9 LS, L2, LC, and L8 loops (middle panel). A portion of the LS loop (left panel) has the most striking differences between haplogroups, and a portion of the LC loop (right panel) contains the second-best region for allelic differences.

including a tyrosine involved in the secondary coordination sphere and other conserved hydrogen-bonding residues (*Span et al., 2017*). Of note, the regions that have the most inter-haplogroup PMO domain differences corresponded to the AA9 nomenclature of LC, LS, and L2 loops (*Danneels et al., 2018*; *Liu et al., 2018*), which have been shown to play a role in substrate binding and specificity. There were also minor differences in the L8 loop located opposite the substrate-binding surface of the PMO domain (*Figure 2B*). The differences between the different haplogroup PMO domains were more apparent when a homology model was constructed using the crystal structure of *Ao*AA11 (PDB: 4MAI) as a template (*Figure 2C*).

It is possible that the product of CWR-1 from each of the haplogroups was either generating a different product profile or recognized a different substrate. Mutations along substrate binding loops can change the regioselectivity of PMOs (C1 vs. C4) partially or entirely, and these residues can determine soluble substrate preferences (*Courtade et al., 2018*; *Danneels et al., 2018*; *Liu et al., 2018*; *Vu et al., 2014a*). The two regions that exhibited the most prominent differences were the LC and LS

loops, which was consistent with the hypothesis that a substrate/product(s) could be the signal for the *cwr* allorecognition checkpoint (*Figure 2C*).

Our *in vivo* data indicated that the PMO domain from *cwr-1*<sup>HG1</sup> was sufficient to confer the *cwr* allorecognition checkpoint. We therefore tested whether the full-length CWR-1 protein from each haplogroup was sufficient for fusion arrest and for conferring allelic specificity. Full-length *cwr-1* alleles from each haplogroup were expressed in a *Δcwr-1Δ81Δcwr-2* strain and confronted against *cwr-1*<sup>HG1</sup> *cwr-2*<sup>HG1</sup> germlings. As expected, strains bearing *cwr-1* from any of the five different haplogroups showed significantly lower fusion rates as compared to a strain carrying *cwr-1*<sup>HG1</sup> and paired with *cwr-1*<sup>HG1</sup> *cwr-2*<sup>HG1</sup> germlings (*Figure 3A and D*). As a control to ensure fusion machinery is not impaired, strains bearing the six different *cwr-1* haplotypes showed high fusion rates with the *Δcwr-1Δ81Δcwr-2* strain (*Figure 3—figure supplement 1A and C*).

To determine whether the PMO domain from the CWR-1 proteins from all six different haplogroups was essential for allorecognition, chimeric constructs containing the N-terminal PMO domain from each HG (2–5) were fused to the GS linker region and X278 domain of *cwr-1*<sup>HG1</sup> (*Figure 3B*) in a *Δcwr-1Δ81Δcwr-2* strain (which also expressed cytoplasmic GFP). The resulting germlings were paired with a *cwr-1*<sup>HG1</sup> *cwr-2*<sup>HG1</sup> strain, a *Δcwr-1Δ81Δcwr-2* mutant, and a *Δcwr-1 cwr-2*<sup>HG1</sup> strain. The fusion percentages between germlings carrying any of the HG2-5 *cwr-1* chimeric constructs were low when paired with a *cwr-1*<sup>HG1</sup> *cwr-2*<sup>HG1</sup> strain or with the *Δcwr-1 cwr-2*<sup>HG1</sup> strain, but were high when paired with the permissive *Δcwr-1Δ81Δcwr-2* mutant (*Figure 3C–F*, *Figure 3—figure supplement 1B and C*). No significant differences were observed in percentages of fusion when either full-length *cwr-1* from the different haplogroups or with chimeric *cwr-1* alleles with the PMO domain from the different haplogroups were paired with a *cwr-1*<sup>HG1</sup> *cwr-2*<sup>HG1</sup> germlings (*Figure 3D*). These experiments showed that allorecognition is mediated in *trans* between a cell containing *cwr-1* and a cell containing *cwr-2* from the different haplogroups and provide strong evidence that the PMO domain from the six different haplogroups conferred the allorecognition fusion checkpoint.

## PMO catalytic activity is not required for the allorecognition fusion checkpoint

Our data above indicated that the PMO domain from each CWR-1 haplogroup was required to confer the allorecognition fusion checkpoint. These data suggested that the CWR-1 PMO domain from each haplogroup might generate a unique product distribution that would confer allorecognition. To test this hypothesis, the PMO domain from the remaining five CWR-1 haplogroups was expressed in a heterologous system, and the protein was purified using the same strategy for the expression of the PMO domain from the HG1 strain (FGSC2489) (*Figure 1D and E*). The CWR-1 proteins from HG2-6 ran slightly high (approximately 1–2 kDa higher than expected) on SDS-PAGE (*Figure 1—figure supplement 4A*) and were confirmed to be the correct mass (*Figure 1—figure supplement 4B*), where each deconvoluted mass was consistent with three disulfides present on the protein (±1 Da) similar to the HG1 protein. All six PMOs from the different haplogroups oxidatively degraded both the α- and β- alloforms of chitin (*Figure 4A and B*), with β-chitin producing more oxidized fragments than α-chitin. Additionally, each of the six CWR-1 PMO proteins from the six different haplogroups generated C1-oxidized products, bound ~1 equivalent of copper and reduced oxygen at similar rates (*Figure 4C and D*). None of the other substrates tested showed activity (*Supplementary file 1f*).

Our *in vitro* data showed no selectivity differences between the CWR-1 PMOs from the six different haplogroups on α- and β-chitin alloforms. We therefore tested whether these HG1-6 PMOs had differences in activity on a more physiologically relevant substrate, the cell wall from an HG1 strain (FGSC2489). This cell wall was purified and assessed as a substrate for the PMO domains from the six different CWR-1 haplogroups. New peaks that corresponded to C1-oxidized chitooligosaccharides and soluble chitooligosaccharides appeared only in the presence of both reductant and the PMO. The products from the CWR-1 PMOs from haplogroups 1,4 and 5 showed a similar pattern of four peaks in the C1-oxidized chitin fragments region (*Figure 4E*), whereas the CWR-1 PMO from haplogroups 2, 3, and 6 showed only two prominent C1-oxidized peaks (*Figure 4E*). Since this pattern did not correlate with phenotypic aspects of cell fusion incompatibility for the six CWR-1 haplogroups, we attributed this result to small rate differences between the CWR-1 PMO domains from the different haplogroups and heterogeneity in the purified cell wall suspension. As a control, cell wall components from the *Δcwr-1Δ81Δcwr-2* strain were purified and used as a substrate. Although all six CWR-1 PMO

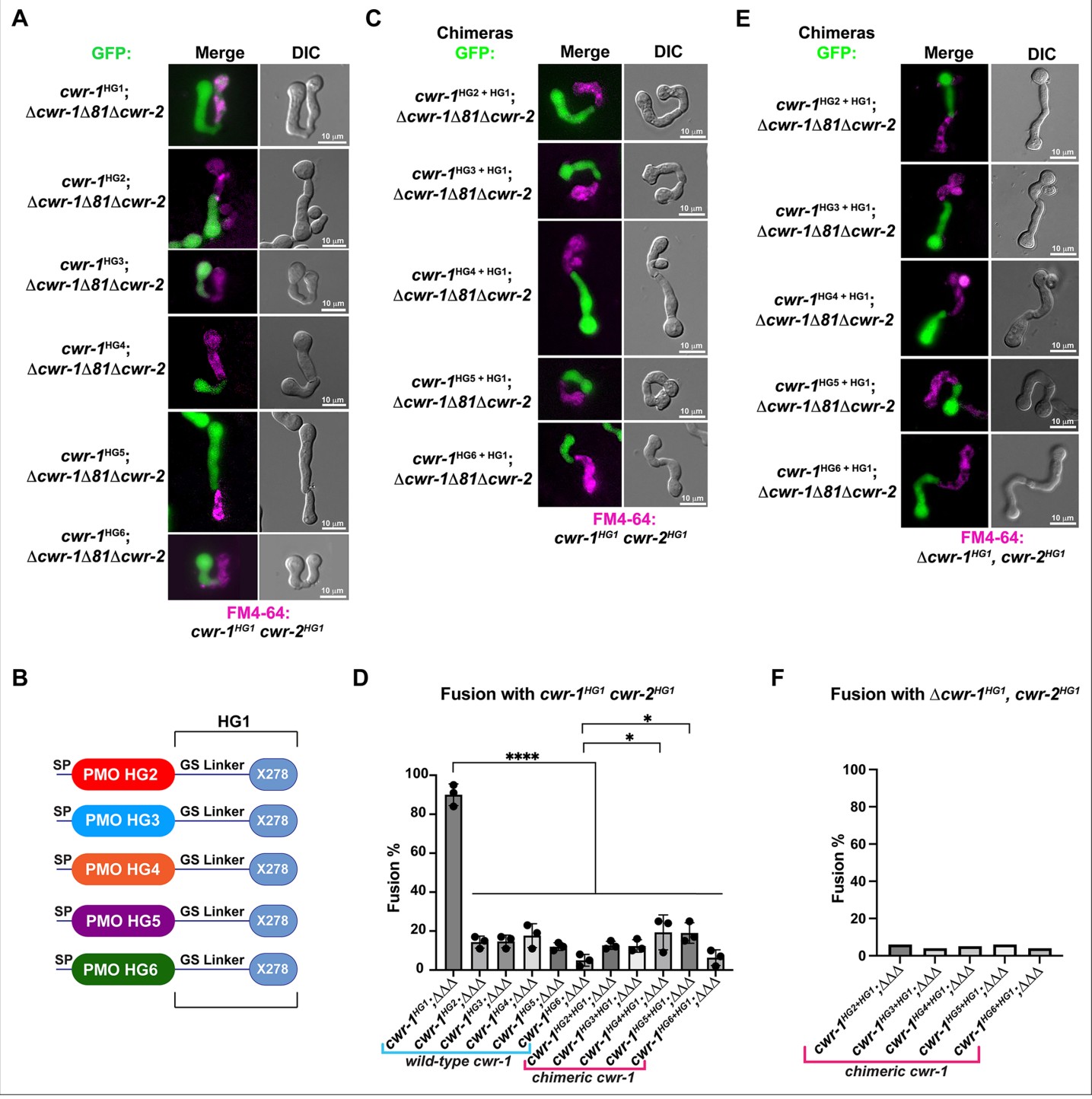

**Figure 3.** The polysaccharide monooxygenase (PMO) domain in CWR-1 functions to confer the allorecognition fusion checkpoint. (**A**) Micrographs of the dominant fusion events between germlings expressing *cwr-1* alleles from each haplogroup in a *Δcwr-1Δ81Δcwr-2* GFP background when paired with *cwr-1^HG1^ cwr-2^HG1^* germlings (FGSC2489) stained with FM4-64. (**B**) CWR-1 chimeras with the PMO domain from the different *cwr* haplogroups (*cwr-1^HG2^* from D111, *cwr-1^HG3^* from JW258, *cwr-1^HG4^* from JW242, *cwr-1^HG5^* from P4476, and *cwr-1^HG6^* from JW228), with the glycine serine linker and X278 domains from *cwr-1^HG1^* (from FGSC2489) are schematically shown. (**C**) Micrographs of the dominant fusion events between germlings expressing the chimeric CWR-1 proteins paired with *cwr-1^HG1^ cwr-2^HG1^* germlings stained with FM4-64. (**D**) Quantification of cell fusion frequencies depicted in (**A**) and (**C**). The experiments were performed in biological triplicate, assessing fusion of 100 germling pairs for each replicate. A one-way ANOVA followed by Tukey's post-hoc test was used for statistical analysis, error bars represent SD, *p<0.05, ****p<0.0001. Individual p-values are reported in *Figure 3—source data 1*. (**E**) Micrographs of the dominant fusion event between GFP germlings expressing the chimeric CWR-1 proteins paired with a *Δcwr-1 cwr-2^HG1^*

*Figure 3 continued on next page*

*Figure 3 continued*

strain stained with FM4-64. (**F**) Quantification of cell fusion frequencies between 100 germlings depicted in (**E**). See *Figure 3—figure supplement 1* for control experiments.

The online version of this article includes the following source data and figure supplement(s) for figure 3:

**Source data 1.** p-Values. *Figure 3D*.

**Figure supplement 1.** Germling fusion phenotype and fusion percentages in strains bearing *cwr-1* from the different haplogroups and *cwr-1* chimeras.

haplogroups exhibited oxidized C1-oxidized products, there appeared to be two main groupings of peak patterns: CWR-1 from HG1 and 5 contained a third oxidized chitin fragment, while CWR-1 from HGs 2, 3, 4, and 6 showed only two prominent peaks (*Figure 4F*). These data indicated that cell walls from both HG1 and the *Δcwr-1Δ81Δcwr-2* strains were similar and that differences could be attributed to small rate differences or cell wall sample heterogeneity. Efforts to identify low-abundance carbohydrates or other potential products through ESI-MS were unsuccessful.

PMO activity was also determined on purified cell walls from a self-incompatible strain (*cwr-1^HG1^cwr-2^HG1^; cwr-1^HG2^*); this strain shows reduced growth, asexual spore production, and lack of self-fusion (*Gonçalves et al., 2019*). However, the pattern of products on this substrate for all six PMO domain haplogroups had very similar product profiles (*Figure 4G*). The PMO reaction products from CWR-1^HG1^ and CWR-1^HG4^ on chitin and purified cell walls from FGSC2489 (HG1) and JW242 (HG4) strains were also used in the germling fusion assays. However, neither the addition of the PMO reaction products nor the cell walls from the incompatible strains blocked or reduced cell fusion frequencies between compatible strains (*Figure 4—figure supplement 1*).

Since there were no striking differences between CWR-1 PMO haplogroups for either products or substrates, the requirement for catalytic activity in fusion blockage was tested. In *Serratia marcescens* and *Pseudomonas aeruginosa,* alanine variants in the second histidine (corresponding to His78 for CWR-1^HG1^) of the histidine brace in AA10 PMOs (CBP21 and CbpD, respectively) disrupt PMO activity with no oxidized products observed (*Askarian et al., 2021*; *Vaaje-Kolstad et al., 2010*). Therefore, we constructed *cwr-1^HG1^* and *cwr-1^HG6^* variants containing His→Ala substitution of the first histidine (His20), a second set of variants containing a His→Ala substitution at H78 and double His→Ala substitutions (H20A; H78A). Similar to the WT PMOs, the variant proteins' deconvoluted mass was consistent with three disulfides and confirmed the construct was correctly expressed (*Figure 5—figure supplement 1*). These CWR-1 PMO His→Ala variants were assessed for catalytic activity, copper-binding, and/or subsequent hydrogen peroxide generation. All single and double histidine variants of PMO^HG1^, PMO^HG6^, and the Tyr→Ala variant PMO^HG1^ were inactive on β-chitin, where no oxidative activity could be detected (*Figure 5A and B*). PMO^HG1^ H20A and H78A variants and PMO^HG6^ H79A variants retained copper-binding after purification (*Figure 5C*). These variants were functional in an oxygen reduction assay but exhibited slightly lower rates as compared to WT CWR-1 PMO (*Figure 4C*). Both PMO^HG1^ and PMO^HG6^ His→Ala double variants greatly reduced copper-binding and had lower oxygen reduction rates (*Figure 5D*). X-band continuous-wave EPR spectra were acquired for the WT PMO^HG1^ and PMO^H78A^ proteins. Both PMO^HG1^ and its H78A variants bound copper; this active site copper center is in the Cu(II) oxidation state under aerobic conditions. Cu(II) has an S = 1/2 spin state, and EPR is a sensitive probe for the primary coordination sphere at the active site. The EPR spectra of the WT and H78A PMO^HG1^ variant proteins were compared to identify any differences in Cu(II) binding between WT and H78A mutant PMO proteins (*Figure 5—figure supplement 2*). The WT PMO^HG1^ spectrum is consistent with other reported PMOs (*Frandsen et al., 2016*) that contain axial Cu (II) sites with observed g values of g1 = 2.245, g2 = 2.065, and g3 = 2.005 with visible N–hyperfine splitting. The spectrum from the PMO^HG1^ H78A variant is distinct from WT PMO^HG1^, displaying more splitting g2 and g3 values, indicative of a more rhombic coordination environment: g1 = 2.245, g2 = 2.08, and g3 = 2.05. The EPR differences between WT and the PMO H78A variant suggests a significant change to the primary coordination sphere of the Cu(II) ion and is presumably because the rigid histidine brace has been changed and replaced by water molecules. Since the PMO^HG6^ variant proteins were not substantially different from the PMO^HG1^ variant proteins, these results likely represent all the PMO proteins in the six haplogroups.

Previously, we tested whether a tyrosine residue in the PMO^HG1^ domain that is predicted to be important for catalysis was essential for allorecognition in pairings with a wild-type strain (JW199,

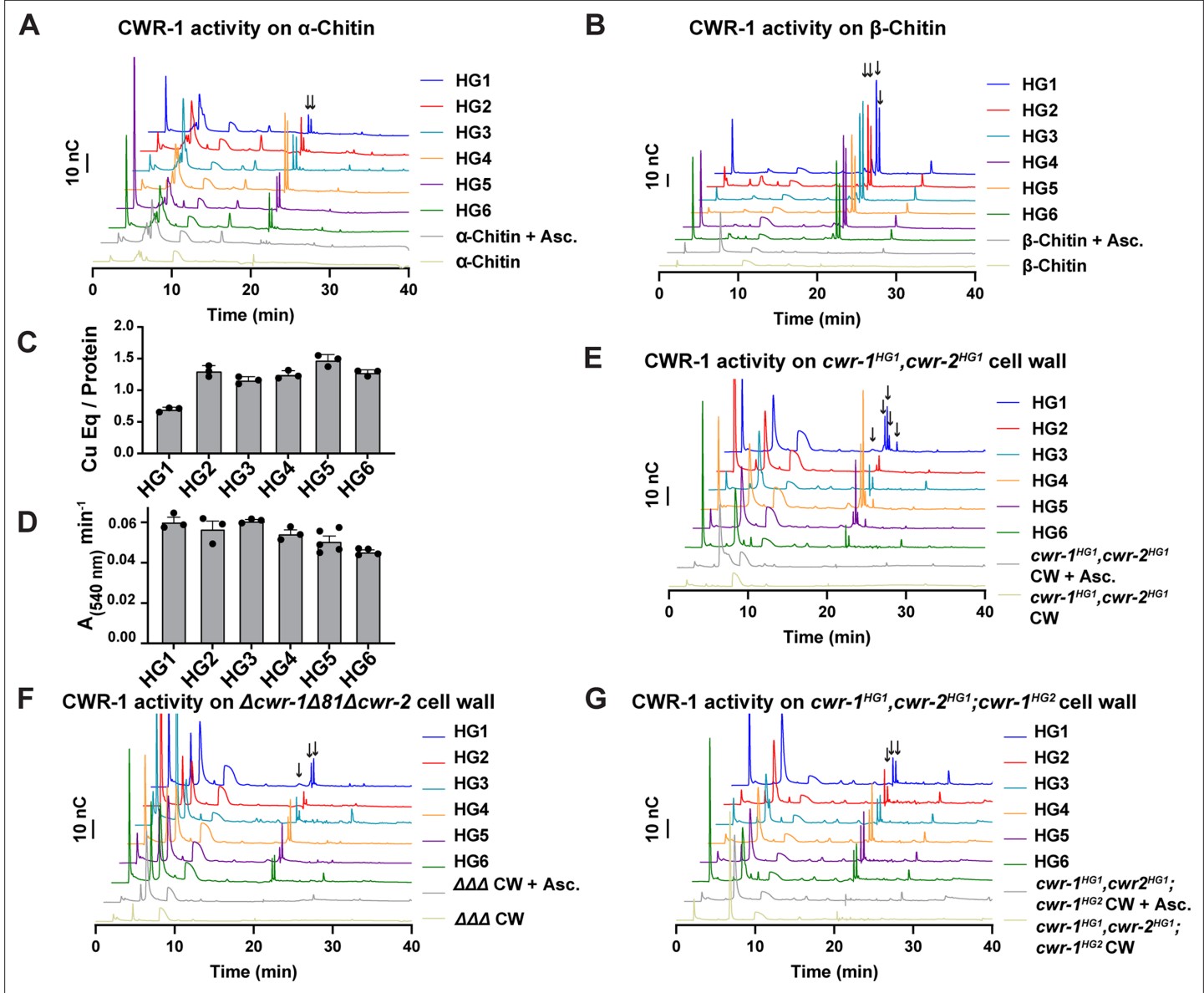

**Figure 4.** Catalytic activity of CWR-1 haplogroups. (**A, B**) A comparison of the reaction products from CWR-1 from each of the six haplogroups on α-chitin and β-chitin. There were minor differences, but each haplogroup generates the same C1-oxidized fragments. (**C, D**) Comparison of bound copper and oxygen reduction of CWR-1 from the six different haplogroups. All of the CWR-1 proteins bound ~1 copper atom per polysaccharide monooxygenase (PMO) and reduced oxygen at similar rates. ICP experiments were done in technical triplicate, and oxygen reduction assays were done with each point representing a biological replicate. All error bars are SEM. Each experiment was performed with an n=3 except for the HG5 and HG6 oxygen reduction assays that have n=5 and n=4, respectively. (**E–G**) A comparison of reaction products of CWR-1 from each haplogroup on purified cell wall from the wild-type HG1 strain (FGSC2489), the *Δcwr-1Δ81Δcwr-2* mutant strain, and a strain expressing a CWR-1[HG2] PMO protein in an HG1 strain (*cwr-1[HG1] cwr-2[HG1]; cwr-1[HG2]*) (*Gonçalves et al., 2019*). There were minor differences between the alleles and substrates, but all contain the same C1-oxidized chitin fragments. Asc. means ascorbate only. Black arrows denote peaks that elute in the region where C1-oxidized products elute. All HPAEC-PAD assays were done in at least biological triplicate. Source data for this figure can be found in *Figure 4—source data 1*, *Figure 4—source data 2*, *Figure 4—source data 3*, *Figure 4—source data 4*, *Figure 4—source data 5*, *Figure 4—source data 6*, and *Figure 4—source data 7*.

The online version of this article includes the following source data and figure supplement(s) for figure 4:

**Source data 1.** HPAEC-PAD source data.

**Source data 2.** HPAEC-PAD source data.

**Source data 3.** ICP source data.

**Source data 4.** Horseradish peroxidase (HRP)-oxygen reduction assay source data.

*Figure 4 continued on next page*

*Figure 4 continued*

**Source data 5.** HPAEC-PAD source data.

**Source data 6.** HPAEC-PAD source data.

**Source data 7.** HPAEC-PAD source data.

**Figure supplement 1.** HPAEC-PAD traces of reaction before fusion experiment and cell fusion percentages.

**Figure supplement 1—source data 1.** HPAEC-PAD source data.

HG5); a statistically significant level of fusion was observed in these pairings, suggesting that Tyr159 was important for allorecognition (*Gonçalves et al., 2019*). Here, we further assessed these findings using an HG1 strain bearing a mutation in *cwr-1* (*cwr-1* $^{HG1\ Y159A}$) in a *Δcwr-1Δ81Δcwr-2* deletion strain and paired with isogenic germlings bearing a *cwr-2*$^{HG3}$ allele in a *Δcwr-1Δ81Δcwr-2* background. In these pairings, a significant block in cell fusion was observed (*Figure 5E and F*). These data suggest that other genetic factors in the JW199 strain affected cell fusion frequencies when paired with the HG1 *cwr-1*$^{Y159A}$ mutant in an otherwise FGSC2489 background. However, using isogenic strains here showed that allorecognition and a significant cell fusion block occurred between a strain bearing only CWR-1$^{Y159A}$ and a strain bearing only CWR-2$^{HG3}$.

Mutations in the histidine brace resulted in chitin-inactive variants, although the purified PMO$^{HG1}$ H20A, PMO$^{HG1}$ H78A, and PMO$^{HG6}$ H79A variants could still bind copper. With the retention of copper, hydrogen peroxide could still be generated from these single His→Ala variants. We, therefore, tested whether the single histidine brace variants (H20A and H78A) and the double His→Ala variant (H20A; H78A) were affected in allorecognition and cell fusion. Isogenic fungal strains were constructed bearing HG1 *cwr-1*$^{H20A}$, *cwr-1*$^{H78A}$, and *cwr-1* $^{H20A;\ H78A}$ alleles. Germlings from these strains were paired with germlings expressing *cwr-2*$^{HG3}$ in the *cwr-1 cwr-2* deletion strain (*cwr-2*$^{HG3}$; *Δcwr-1Δ81Δcwr-2*) that also expressed cytoplasmic GFP. Contrary to expectations, the single *cwr-1* histidine variants (H20A and H78A), including the inactive double mutant (H20A; H78A), still retained the ability to trigger allorecognition and block cell fusion when paired with germlings that expressed an incompatible *cwr-2*$^{HG3}$ allele (*Figure 5E and F*). As a control, all the His→Ala *cwr-1*$^{HG1}$ variants were paired with *cwr-1*$^{HG1}$ *cwr-2*$^{HG1}$ (FGSC2489) germlings, and all underwent robust cell fusion (*Figure 1—figure supplement 3C and D*). These data rule out hydrogen peroxide generation as a contributing factor in blocking cell fusion as the copper-deficient *cwr-1*$^{H20A;\ H78A}$ variant was not different from those that retained some copper-binding. Taken together, these data suggest that the *cwr* fusion block checkpoint is not dependent on enzymatic activity, and the CWR-1 PMO domain is not generating a signal to block cell fusion via a substrate preference or generated product.

## PMO domain loops confer specificity for cell fusion block

Our data showed that the PMO domain is essential for allorecognition, but that catalytic activity of the PMO domain was not required. An examination of a homology model of the six different CWR-1 PMO haplogroups based on the structure of *Ao*AA11 revealed that differences appeared to be most significant in the region of the LS loop (*Figure 2C*). To assess whether this region of CWR-1 was important for conferring allorecognition specificity, we constructed a loop-swap chimeric construct: the first where the entire LS loop and surrounding amino acids from PMO$^{HG1}$ (I85 to G129) was replaced with the sequence from PMO$^{HG6}$ (V86 to T130) (*Figure 6A*). If this region of the CWR-1 protein was necessary for allelic specificity, the chimeric strains would have a higher fusion frequency with a *cwr-2*$^{HG6}$ (in a *Δcwr-1Δ81Δcwr-2* background) strain and a lower fusion frequency with the *cwr-1*$^{HG1}$ *cwr-2*$^{HG1}$ strain or *cwr-2*$^{HG1}$ in the *Δcwr-1Δ81Δcwr-2* background. However, the *cwr-1*$^{HG1}$LS $^{HG6}$ chimera maintained the same specificity as HG1 strain (*Figure 6B and C*). These studies indicate that this region, which is different in all six CWR-1 PMO haplogroups, was insufficient to switch allelic specificity. A second chimeric construct was generated; this new chimera includes LS loop, L8 loop, and the first half of the LC loop. The L8 and LC regions showed differences between haplogroups in the MSA and homology models, but more so in the case of the LC region. We hypothesized that a combination of these three regions could impart specificity. The region I85 to D201 of CWR-1$^{HG1}$ was replaced with V86-D202 from CWR-1$^{HG6}$ (*Figure 6A*). This *cwr-1*$^{HG1/HG6}$ chimera showed increased fusion percentages (~55%) with *cwr-2*$^{HG6}$ cells, but a decreased fusion percentage (~35%) with both the HG1 strain (FGSC2489) and a *cwr-2*$^{HG1}$ strain in the *Δcwr-1Δ81Δcwr-2* background (*Figure 6D and E*). These results suggest

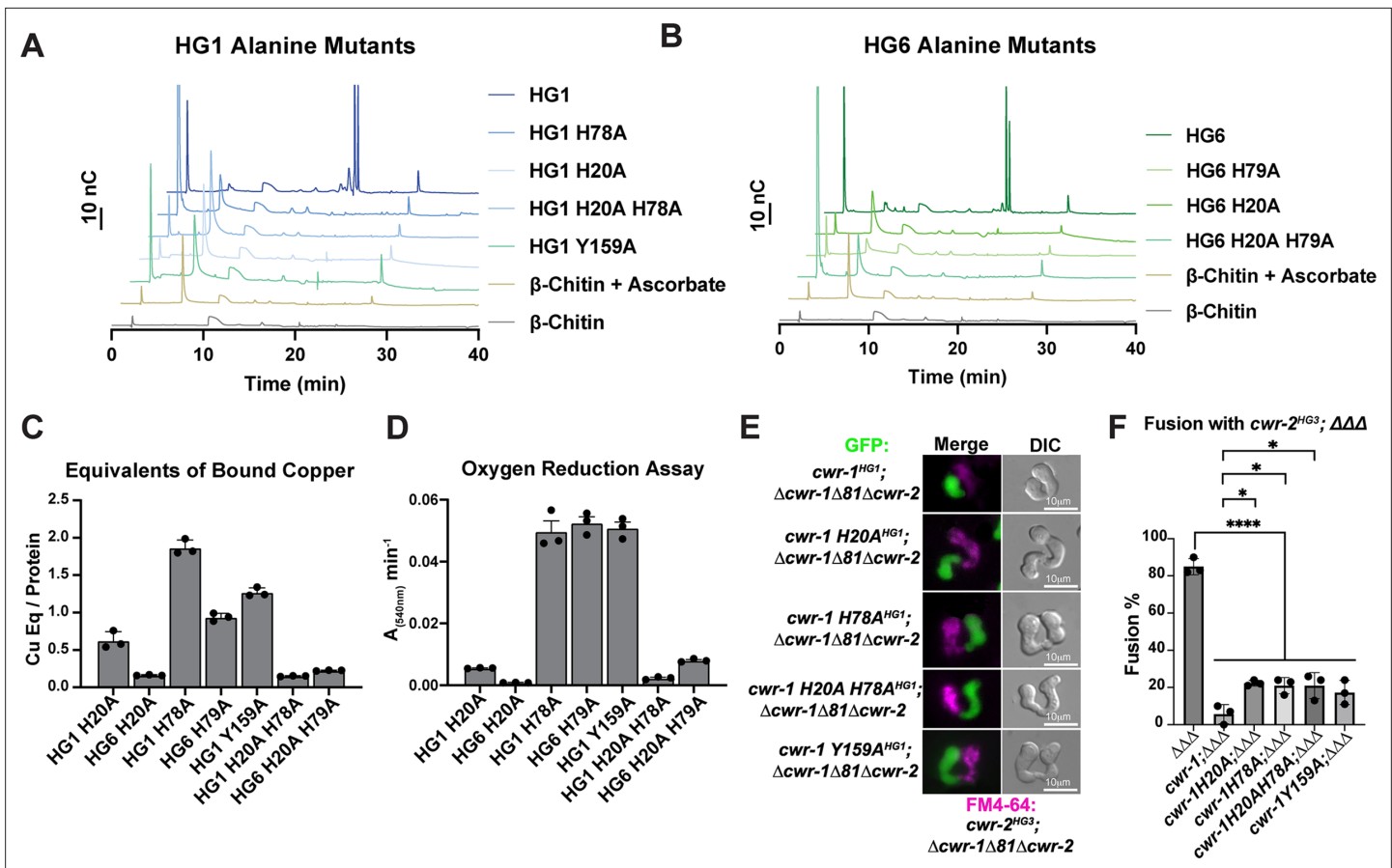

**Figure 5.** Characterization of CWR-1 polysaccharide monooxygenase (PMO) active-site variants. (**A, B**) Comparison of the ability of active-site variants of the PMO[HG1] and PMO[HG6] domains and their ability to degrade β-chitin. Mutation of one or both of the histidine brace residues abolished chitin-oxidizing activity. All HPAEC-PAD assays were done in at least biological triplicate. (**C**) Comparison of bound copper of active-site variants. Mutating the second histidine alone did not abolish copper-binding; however, mutation of the first histidine dramatically reduced the amount of copper bound, and mutation of both histidine residues almost eliminated copper-binding entirely. ICP experiments were performed in technical triplicate. Error bars represent SEM. (**D**) Comparison of oxygen-reduction activity of active site mutants. All constructs that contained the first histidine to alanine variant had significantly reduced oxygen reduction activity. Each data point represents a biological replicate. Error bars represent SEM. (**E**) Fusion test of the different CWR-1[HG1] histidine variants and tyrosine variant paired with a strain expressing *cwr-2[HG3]* in the Δ*cwr-1Δ81Δcwr-2* mutant background. (**F**) Quantification of cell fusion percentages depicted in (**E**). The experiments were performed in biological triplicate, counting 100 germling pairs for each replicate. A one-way ANOVA followed by Tukey's post-hoc test was used for statistical analysis, error bars represent SD, *p<0.05, ****p<0.0001. Individual p-values are reported in *Figure 5—source data 5*. The fusion percentages with *ΔΔΔ* and *cwr-1[HG1]; ΔΔΔ* are the same as in *Figure 1C*. Source data for this figure can be found in *Figure 5—source data 1*, *Figure 5—source data 2*, *Figure 5—source data 3*, and *Figure 5—source data 4*.

The online version of this article includes the following source data and figure supplement(s) for figure 5:

**Source data 1.** HPAEC-PAD source data.

**Source data 2.** HPAEC-PAD source data.

**Source data 3.** ICP source data.

**Source data 4.** Horseradish peroxidase (HRP)-oxygen reduction assay source data.

**Source data 5.** p-Values.

**Figure supplement 1.** Deconvoluted mass spectrum of polysaccharide monooxygenase (PMO) histidine variants.

**Figure supplement 1—source data 1.** Whole-protein MS data.

**Figure supplement 2.** X-band EPR and simulated spectra of PMO[HG1] and PMO[HG1] H78A variant.

**Figure supplement 2—source data 1.** EPR source data.

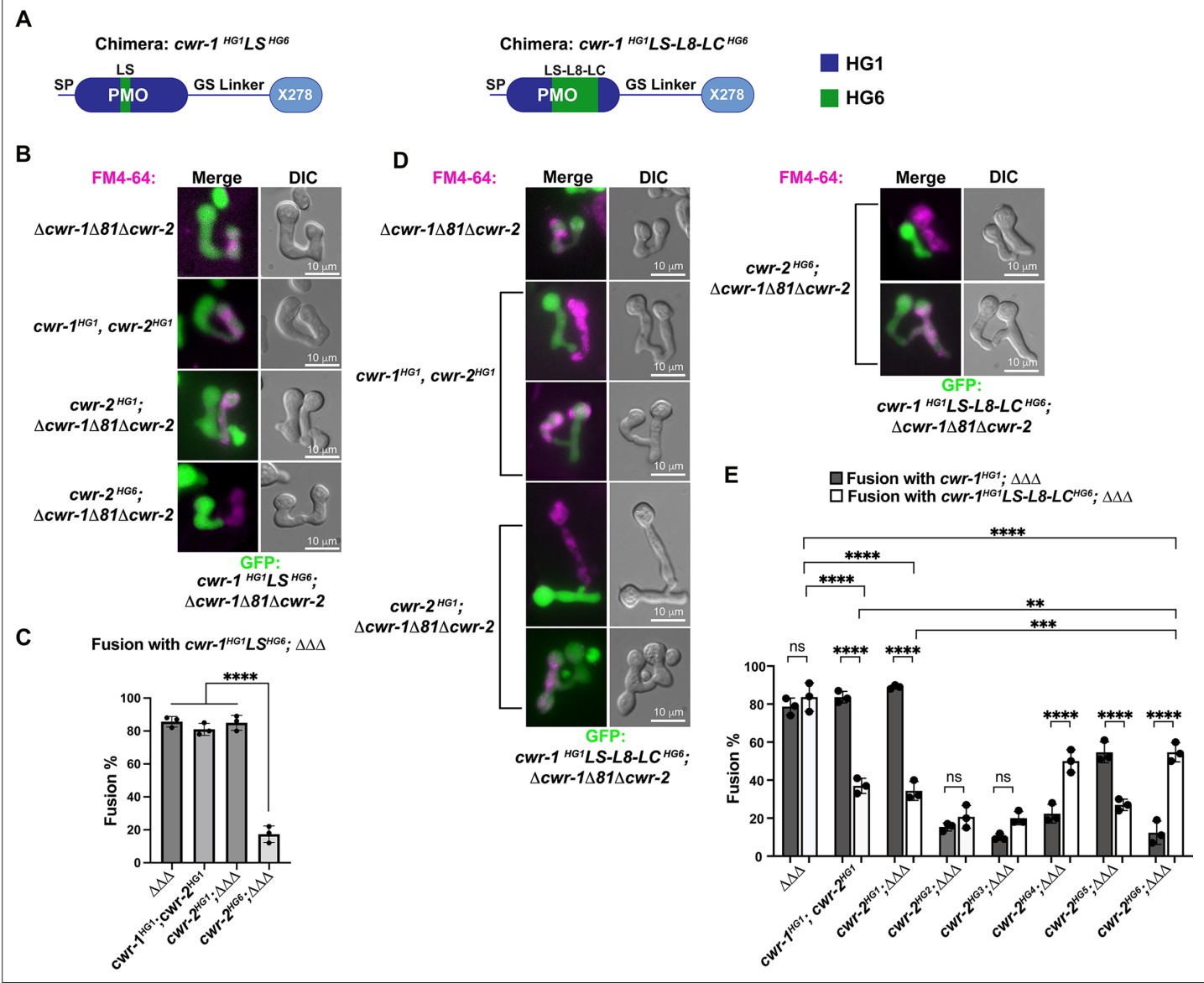

**Figure 6.** CWR-1 chimeras to define the polysaccharide monooxygenase (PMO) haplogroup specificity region. (**A**) A schematic depicting where the loops from PMO$^{HG1}$ domain that were replaced with loops from PMO$^{HG6}$ (green). SP means signal peptide; GS linker means the glycine/serine-rich region that connects the catalytic domain to the X278, all of which were derived from CWR-1$^{HG1}$. Navy is the region of the protein that is from PMO$^{HG1}$, green is the region of the protein from PMO$^{HG6}$. (**B**) Cells expressing the chimera *cwr-1$^{HG1}$LS$^{HG6 (V86-T130)}$* in a *Δcwr-1Δ81Δcwr-2* GFP strain were paired with indicated FM4-64 stained germlings. (**C**) Cell fusion percentages between germlings depicted in (**B**). The experiments were performed in biological triplicate, counting 100 germling pairs for each replicate. For statistical analysis, a one-way ANOVA followed by Tukey's post-hoc test was used, error bars represent SD, ****p<0.0001. Individual p-values are reported in *Figure 6—source data 1*. (**D**) Cells expressing the chimera *cwr-1$^{HG1}$LS-L8-LC$^{HG6}$* $^{(V86-D202)}$ in a *Δcwr-1Δ81Δcwr-2* GFP strain were paired with indicated FM4-64-stained germlings, with examples of blocked and fusing cells. (**E**) Cell fusion percentages between the chimeric strain (*cwr-1$^{HG1}$LS-L8-LC$^{HG6}$*) and strains harboring the *cwr-2* alleles from all six haplogroups. Cell fusion percentages between *cwr-1$^{HG1}$* cells with strains carrying *cwr-2* alleles from different haplogroups were shown in *Figure 1—figure supplement 3A and B*. The experiments were performed in biological triplicate, counting 100 germling pairs for each replicate. For statistical analysis, a two-way ANOVA followed by Tukey's post-hoc test was used, error bars represent SD, **p<0.01, ***p<0.001, ****p<0.0001, ns, not significant. Individual p-values are reported in *Figure 6—source data 2*.

The online version of this article includes the following source data for figure 6:

**Source data 1.** p-Values.

**Source data 2.** p-Values.

that the V86-D202 region in CWR-1 contains some but not all required residues to elicit the allorecognition response. To determine whether we had created an allele with a new *cwr-1* specificity, we assessed fusion percentages of this chimera with strains bearing the other *cwr-2* haplogroup alleles in a Δ*cwr-1*Δ*81*Δ*cwr-2* background. The *cwr-1*$^{HG1/HG6}$ chimeric strain showed fusion percentages of ~21, ~20, and ~27% with strains bearing *cwr-2*$^{HG2}$, *cwr-2*$^{HG3}$, and *cwr-2*$^{HG5}$, but a higher fusion percentage with a *cwr-2*$^{HG4}$ strain (~50%) (**Figure 6E**). Consistent with alterations in allelic specificity, these results were different from the *cwr-1*$^{HG1}$ germlings paired with *cwr-2* germlings from the different haplogroups (**Figure 6E**). Thus, strains with the *cwr-1*$^{HG1/HG6}$ chimera showed the highest cell fusion percentages with the *cwr-2*$^{HG4}$ and *cwr-2*$^{HG6}$ strains but were not fully compatible with any of the strains containing *cwr-2* from the different haplogroups. These data and further studies into chimeric constructs of *cwr-1* in concert with chimeric constructs of *cwr-2* will help to define specific residues/regions required for haplogroup specificity and allorecognition that will enable the development of molecular models for somatic cell fusion arrest via the cell wall remodeling checkpoint.

## Discussion

The *N. crassa* cell wall remodeling checkpoint acts in *trans* via interactions between incompatible CWR-1 and CWR-2 during the process of cell fusion. If germlings or hyphae bearing *cwr-1* and *cwr-2* alleles of identical haplogroup specificity contact each other following chemotropic growth, cell wall

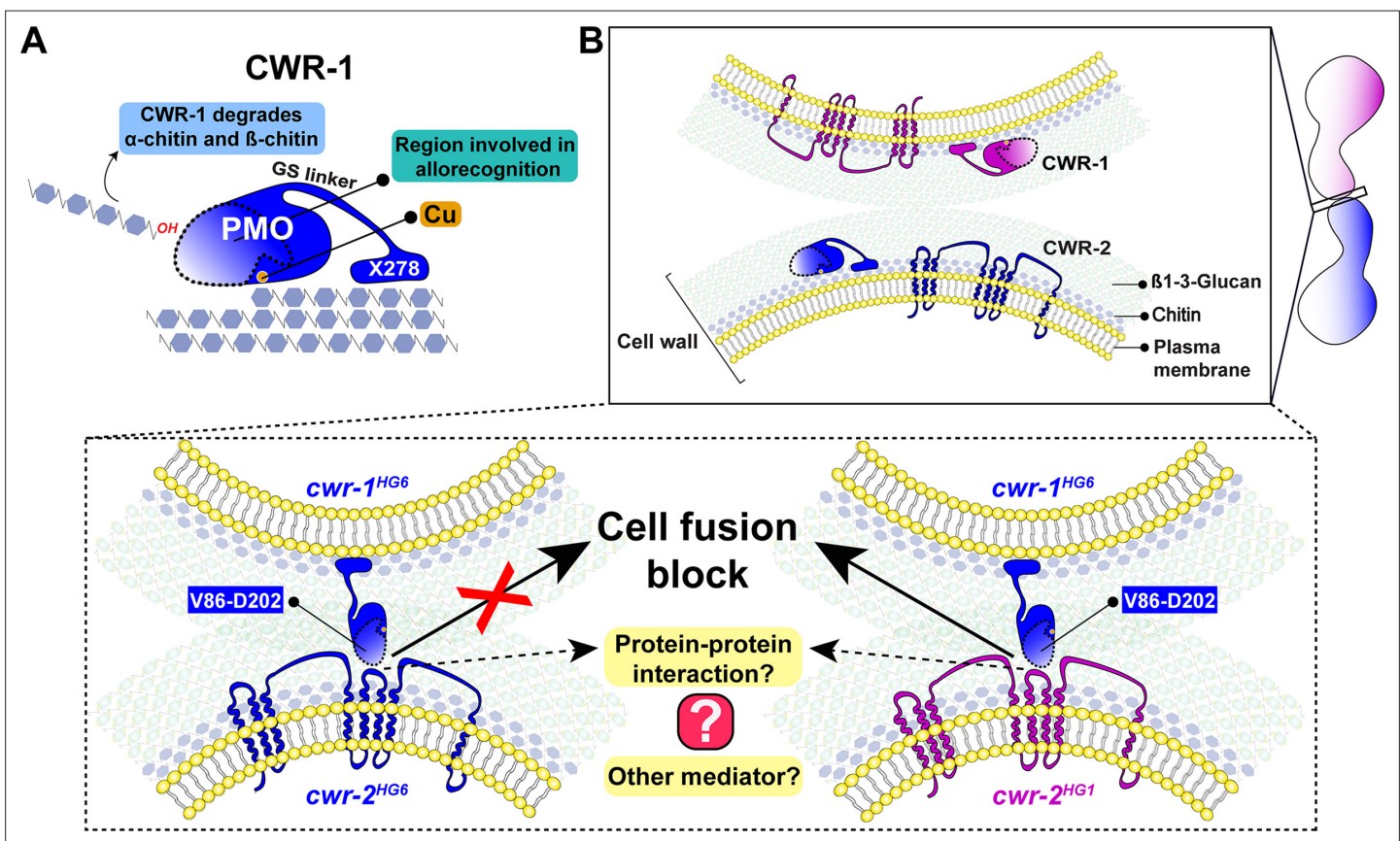

**Figure 7.** A schematic model that summarizes the role of CWR-1 and CWR-2 in allorecognition at the cell fusion checkpoint. (**A**) The polysaccharide monooxygenase (PMO) domain, a glycine and serine-rich region, and a putative chitin-binding module, X278, of CWR-1 are shown. The PMO domain degrades chitin, and the region important for allorecognition is highlighted. (**B**) The top panel shows a schematic showing the approach of the two tips of germlings before cell–cell contact. Once the germlings approach each other, the bottom two panels show possible outcomes. In the left panel, both germlings express a CWR-1 and CWR-2 protein belonging to the same haplogroup. This pair of germlings will not trigger the cell fusion block signal and will undergo cell fusion. In the right panel, the germlings express incompatible CWR-1 and CWR-2 proteins from different haplogroups, thereby eliciting a cell-fusion-blocking signal. The LC-LS-L8 (V86-D202) region of the protein is represented by the area inside the dotted line. It remains to be answered how CWR-1 and CWR-2 interact either directly through a protein–protein interaction or a mediator and which regions of CWR-2 are involved.

dissolution is triggered, whereas cell fusion arrest occurs if strains carry *cwr-1* and *cwr-2* alleles of different haplogroup specificity. In this study, we analyzed cell fusion capability between engineered strains that were otherwise isogenic and harboring either *cwr-1* or *cwr-2* alleles from the distinct haplogroups or *cwr-1* chimeras. We determined that the PMO domain alone confers allelic specificity. However, our mutational, biochemical, and *cwr-1* chimeric analyses showed that the PMO catalytic activity was not required for allorecognition and cell fusion arrest. These results suggest that CWR-1 and CWR-2 interact nonenzymatically in *trans* to trigger a block in cell wall dissolution. This interaction may include contact point(s) outside of the LS, LC, or L8 region of the CWR-1 PMO domain as the chimeric constructs did not fully switch allelic specificity. This CWR-1 region may interact with the CWR-2 domains that face the cell wall or another, but unidentified, mediating partner may be important (*Figure 7*). Mechanistically, how CWR-1/CWR-2 interactions block in cell fusion is unclear, but we hypothesize that aspects associated with cell wall dissolution, including vesicle delivery or activation of cell wall remodeling enzymes, could potentially be targets.

The CWR-1 PMO domain from all six haplogroups exhibits C1-oxidative activity on chitin and degrades β-chitin more efficiently than the α-chitin alloform. Although the PMO domain is only one part of a multidomain protein (PMO, GS linker, and X278), its activity is similar to the characterized, single-domain AA11 PMO (*Støpamo et al., 2021*). These data suggest that these two types of architectures in the context of chitin degradation may be redundant, that is, both architectures perform the same reaction that has been observed previously for AA9 and AA13 PMOs (*Vu et al., 2019*). The biological role of the long GS linker and carbohydrate-binding module remains unclear. However, this linker and carbohydrate-binding module may allow the CWR-1 protein to remain bound to the cell wall through chitin–protein interactions, which may be important for non-allorecognition functions, including CWR-1 PMO activity. This work shows that CWR-1 can degrade chitin present in the fungal cell wall, which may aid in cell wall dissolution; the *Δcwr-1Δ81Δcwr-2* mutant shows a slight delay in fusion between compatible cells (*Gonçalves et al., 2019*) and is thus consistent with this hypothesis.

In biochemical analyses of the PMO domain catalytic function, it was surprising that several His→Ala variants still bound copper with 1:1 stoichiometry after reconstitution. Even though copper is bound, oxygen-dependent chitin turnover was abolished. The EPR of the bound copper showed the importance of the histidine brace in catalysis. Mutations of either the Cu-coordinating histidine or the non-coordinating axial tyrosine to alanine eliminated all catalytic activity on insoluble substrates. Previously, it was demonstrated that mutation of the axial tyrosine to phenylalanine in an AA10 PMO from *Thermobifida fusca* or an AA9 PMO from *Hypocrea jecorina* resulted in a less, but still active protein that maintains the ability to degrade phosphoric-acid swollen cellulose (PASC) (*Kruer-Zerhusen et al., 2017*; *Jones et al., 2020*). In nature, PMOs without tyrosine in this position often have phenylalanine as a replacement. It may be that any aromatic residue is essential in this position for oxygen-dependent catalysis and has some role in forcing the copper into the correct geometry through longer-range structural interactions.

Peroxide-generating activity was observed in single CWR-1 histidine variants, yet subsequent Fenton-like chemistry that would cleave chitin did not occur. Thus, *in vivo*, the single histidine brace variants may retain the ability to bind copper and subsequently reduce oxygen, and while inactive on carbohydrate substrates, they will reduce $O_2$ to $H_2O_2$ and may elicit an oxidative response. However, the block in cell fusion observed in germlings carrying single histidine brace CWR-1 variants suggests that $H_2O_2$ is not a signal involved in cell fusion. The increase in cell fusion percentages in strains carrying truncated versions of *cwr-1*[HG1], or the histidine or tyrosine variants (~20%) in pairings with *cwr-2* incompatible cells, as compared to fusion percentages in strains with wild-type *cwr-1*[HG1] (~6%) (*Figure 1B and C*, *Figure 5E and F*), may indicate that these changes may affect stability or alter CWR-1 conformational structure. Altogether, our data support that the catalytically dead CWR-1 variants still function to impose the cell wall remodeling checkpoint.

The region of the PMO domain essential for allelic specificity encompasses additional residues than what were swapped in the chimeric-loop constructs. As homology models may be inaccurate, changing alleles may require all three loops: L2, LS, and LC. More advanced structural studies would provide insight into the inter-haplogroup differences between CWR-1 proteins from the different haplogroups. The *cwr-1*[HG1] *LS-L8-LC*[HG6] chimera showed a reduction in cell fusion percentages when paired with strains expressing *cwr-2* from any of the six different haplogroups (*Figure 6E*). These data suggest that the CWR-1[HG1/HG6] chimera expresses a functional protein capable of allorecognition and

that this chimera may confer a novel *cwr-1* specificity. This *cwr-1^{HG1} LS-L8-LC^{HG6}* chimera will be useful in defining domains of CWR-2 that confer haplogroup allelic specificity and raises the possibility of constructing a new *cwr* haplogroup by the construction of a *cwr-2* chimera that is fully compatible with strains carrying the *cwr-1^{HG1} LS-L8-LC^{HG6}* chimera. It should be noted, however, that wild isolates contain both *cwr-1* and *cwr-2* alleles, and thus potentially two incompatible *cwr-1-cwr-2* interactions can be triggered during the cell fusion process.

There are several examples of proteins that have been co-opted by biology to perform a completely different role in the cell. This behavior is termed 'moonlighting' (*Jeffery, 1999*; *Jeffery, 2018*) and includes enzymes that form structural components of mammalian eyeball lenses (*Piatigorsky, 1998*), an aconitase that acts as an iron sensor (*Gunawardena et al., 2016*). There has also been a recently characterized chitin-degrading PMO from *P. aeruginosa*, CopD, that is involved in immune evasion. However, the CopD PMO active site appears to be essential for this effect as histidine to alanine variants disrupt the immune evasion phenotype and therefore would not be considered a true moonlighting function (*Askarian et al., 2021*).

Since three of the four characterized AA11s are so similar in both sequence and architecture (*Figure 1—figure supplements 1 and 2*), it may be the case that these three characterized AA11s are also functioning in the same biological role as CWR-1. RNA-seq experiments show *cwr-1* transcripts are upregulated during starvation (*Wu et al., 2020*) and fruiting body development (*Zheng et al., 2014*), which may suggest that *cwr-1* has two biological roles. Allorecognition loci in filamentous fungi are under balancing selection, with defined haplogroups in population samples (*Gonçalves et al., 2020*). Since many fungal species do not have multiple genomes sequenced and available in online databases, and many others do not have facile genetic methods, it is unclear how widespread the role of *cwr-1/cwr-2* is in allorecognition. However, there are several other species that have linked *cwr-1/cwr-2* loci that fall into haplogroups in population samples, including *Fusarium verticillioides, Fusarium fujikuroi, Trichoderma harzianum, Zymoseptoria tritici,* and *Fusarium oxysporum* (*Gonçalves et al., 2019*). Future work will define regions of allelic specificity in CWR-1 and corresponding regions of CWR-2 and probe mechanistic aspects of how the interactions/functions of these proteins mediate the cell wall remodeling checkpoint.

## Materials and methods

### Recombinant DNA techniques and plasmid constructs

Plasmids and oligonucleotides used in this work are listed in *Supplementary file 1a–c*. All plasmids used to transform *N. crassa* were derived from plasmid pMF272 (*Freitag et al., 2004*). PCR reactions were performed in MiniAmp Plus Thermal Cycler with Q5 High-Fidelity DNA polymerase (2000 U/mL) (NEB, Ipswich, MA) according to the manufacturer's instructions. Genes were amplified using genomic DNA of *N. crassa* strains (FGSC2489 or P4471, JW228, JW242, JW258, P4476, D111) as a template. The fragments amplified were purified using Monarch DNA Gel extraction Kit (T1020S, NEB). The *cwr-1* chimeras were made by fusing two fragments, a fragment that encodes the signal peptide and the PMO domain corresponding to alleles from each of the six haplogroups of *cwr-1* (JW228, JW242, JW258, P4471, P4476, D111) and a second fragment that included the region corresponding to the GS linker and X278 domain of *cwr-1^{FGSC2489}* (haplogroup 1). The fragments were fused using polymerase Phusion high-fidelity DNA Polymerase (M0530S, NEB) following the manufacturer's instructions. We designed primers flanking the ORF of the *cwr-1* (NCU01380), *cwr-2* (NCU01382), and chimeras of *cwr-1* with *Xba*I and *Pac*I restriction sites for the 5′ and 3′ ends, respectively. Amplified and gel-purified PCR ORFs were *Xba*I (R0145S, NEB) and *Pac*I (R0547S, NEB) digested and inserted into *Xba*I/*Pac*I digested *pMF272::his-3-Ptef*-1-V5-T*ccg-1*. The fragments were gel purified and ligated using T4 DNA ligase (15224-017, Invitrogen Life Technologies, Waltham, MA). The resulting constructs were used to transform *N. crassa* (△NCU01380△NCU01381△NCU01382::*hph*; *his-3*).

The plasmids to transform *N. crassa* with the truncated version of *cwr-1* (*cwr-1△GS^{HG1}*, *cwr-1△GS△X278^{HG1}*), histidine mutants (*cwr-1H20A^{HG1}*, *cwr-1H78A^{HG1}*, *cwr-1H20AH78A^{HG1}*), and the loop chimera (*cwr-1^{HG1}LS^{HG6(V86-T130)}*) were constructed using the plasmid *pMF272::his-3-Pcwr-1-cwr-1(NCU01380)-Tcwr-1^{FGSC2489HG1}* as a backbone. According to the manufacturer's instructions, the truncated versions of *cwr-1* and the histidine mutant plasmids were prepared using the Q5 Site-Directed Mutagenesis Kit (E0552S, NEB). Gibson assembly (E2611S, NEB) cloning was used for both the

loop chimera constructs and the pET22B PMO domain protein-expression constructs (for all alleles) according to the manufacturer's instructions.

The generated plasmids were transformed into NEB 5-alpha Competent *E. coli* (NEB #C2987) for propagation and storage. Plasmid DNA was extracted with QIAprep Spin Miniprep Kit (QIAGEN, Redwood City, CA; Cat# 27104) and linearized with *Nde*I (NEB, R0111S), *Ssp*I-HF (R3132S), or *Pci*I (NEB, R0655S) to transform *N. crassa*.

## Strain and culture conditions

The standard protocols for *N. crassa* growth, crosses, and maintenance are available at the Fungal Genetics Stock Center (FGSC) website (http://www.fgsc.net/Neurospora/NeurosporaProtocolGuide. htm; accessed 1/23/2019). All the strains used in this work are listed in *Supplementary file 1d* and are available from the FGSC. The strains were grown in Vogel's minimal medium (VMM) (*Vogel, 1956*) solidified with 1.5% agar with supplements as required. For crosses, Westergaard's synthetic cross-medium was used (*Westergaard and Mitchell, 1947*). The FGSC2489 laboratory strain was used as a parental strain and as a WT control for all experiments. The wild *N. crassa* strains used in this study were isolated from Louisiana, USA (*Gonçalves et al., 2019*; *Heller et al., 2016*; *McCluskey et al., 2010*; *Palma-Guerrero et al., 2013*).

## Transformation of *N. crassa*

Conidia of the auxotrophic histidine strain in the triple delete background (Δ*NCU01381*Δ*NCU01381N-CU01382::hph; his-3*⁻) (*Gonçalves et al., 2019*) of *N. crassa* were transformed with linearized plasmid or PCR products by electroporation on a Bio-Rad Pulse controller plus and Bio-Rad gene pulser II. The conidia were electroporated using 1 mm gap cuvettes (Bio-Rad Gene Pulser/MicroPulser Cuvette; Bio-Rad, Hercules, CA) at 1.5 kV, 600 ohm, 25 µF as previously described (*Margolin et al., 1997*). For each transformation, around 30 (His⁺) prototroph transformants were selected and transferred to tubes containing VMM with Hygromycin B. The integration of the constructs in the selected transformants was confirmed by using F-130WH Phire Plant Direct PCR Kit (Thermo Fisher Scientific, Waltham, MA) and Sanger sequencing. Hygromycin B (10687010, 50 mg/mL; Thermo Fisher Scientific) was used at a final 200 µg/mL concentration for selection of transformants. Cyclosporin A (30024, Sigma, St. Louis, MO) was used at a final concentration of 5 µg/mL for selection. The L-histidine (L-histidine hydrochloride monohydrate, 98%, Acros Organics, Waltham, MA) was used at a final concentration of 0.5 mg/mL to support the growth of his-auxotrophic strains.

## *N. crassa* crosses

Heterokaryotic transformants were used as a female parent strain and were grown on Westergaard's synthetic cross-medium at 25°C with constant light until protoperithecia were observed. The protoperithecia were fertilized with a conidial suspension from a male parental strain, FGSC9716/his⁻ or FGSC2489::GFP/his⁻ (FGSC2489; *csr-1::Pccg-1-gfp; his-3*⁻), and grown for an additional 10 days or until ascospores had been shot onto the lid of the Petri dish. To select homokaryons, the resulting ascospores were heat-shocked at 60°C for 40 min and inoculated onto bottom agar plates (VMM with 1.5% agar and a mixture of 20% sorbose, 0.5% fructose, 0.5% glucose) and incubated at 30°C overnight. The germinated ascospores were dissected using a stereomicroscope Stemi SV6 Zeiss and transferred to slants with VMM. The selection of the homokaryons was made depending on their (His⁺) prototrophy, Hygromycin B (Hyg⁺), and/or Cyclosporin A (Cyclosporin⁺) resistance. Additionally, PCR was done to confirm the presence of the construct using F-130WH Phire Plant Direct PCR Kit (Thermo Fisher Scientific) and Sanger sequencing.

## Germling-fusion assays

The strains were grown in VMM in slant tubes for 4 days at 30°C; then, the tubes were placed at room temperature (RT) with constant light for 2 days. The conidia were resuspended in 2 mL of sterile ddH₂O and filtered using cheesecloth. The styryl dye N-(3-triethylammoniumpropyl)–4-(6-(4-(diethylamino) phenyl) hexatrienyl) pyridinium dibromide (FM4-64, 514/670 nm absorption/emission Invitrogen, Waltham, MA) was used at a final concentration of 16.5 µM in ddH₂O from a 16.5 mM stock solution in DMSO, to stain germlings of *N. crassa*. An aliquot of 200 µL of the filtered conidia was stained with 40 µL of FM4-64 (16.5 µM) and incubated at RT for 15 min in the

dark. The stained conidia were centrifuged in a microcentrifuge at 5000 rpm for 2 min, the supernatant was removed, and the conidia were washed twice with 1 mL of sterile ddH$_2$O. The conidia were resuspended in 100 µL of sterile ddH$_2$O and counted using the hemocytometer. The conidial suspension was adjusted to $3 \times 10^7$ conidia/mL. An aliquot of 45 µL of the conidial suspension from the strain stained with FM4-64 was mixed with 45 µL of a conidial suspension from the strain that expresses cytoplasmic GFP. Then, 80 µL of the final mixed suspension was spread on VMM agar plates (60 mm × 15 mm). Plates were incubated for 3.5 hr at 30°C. Agar rectangles of 3 cm × 2 cm approximately were excised and observed with a microscope ZEISS Axioskop 2 MOT. The microscope is equipped with an Illuminator Cool LED microscope, simply Better Control pE-300 white. The images were captured with a Q IMAGING FAST 1394 COOLED MONO 12 BIT microscope camera RETIGA 2000R SN:Q31594 (01-RET-2000R-F-M-12-C) viewed through a Ph3 ×40/1.30 ∞/0.17 Plan-Neofluar oil immersion objective and then processed using iVision-Mac Scientific Image Processing Bio Vision Technologies (iVision 4.5.6r4). The fusion percentage of germlings was evaluated through observation of cytoplasmic mixing of GFP into germlings stained with FM4-64. Pictures for at least 15 fields were taken (for each filter DIC, Blue, GreenRed), depicting a total of at least 100 germling-contact events, with three biological replicates. Quantification is shown with error bars as SD. All the images were further processed using Fiji (ImageJ2, version 2.3.0/1.53f; *Schneider et al., 2012*), Adobe Photoshop 2022 version 23.1.0, and Adobe Illustrator 2022 version 26.0.2. ANOVA statistical analyses were performed using GraphPad Prism 9. ANOVA statistical analysis was selected to assess differences in data between different strains/genotypes. One-way ANOVA for the analysis that involves one independent variable and two-way ANOVA for analysis with two independent variables was followed by Tukey's test to perform multiple comparisons to assess significant differences. All p-values and 95% CIs for each analysis are reported in p-values source data.

## CWR-1 catalytic domain expression

Each *cwr*-1 allele was amplified from genomic DNA and cloned into the pet22B plasmid containing the pelB leader sequence. Each construct contained the PMO catalytic domain of the *cwr-1* gene with the signal peptide removed, followed by a C-terminal hexa-histidine tag with a thrombin cleavage site. The constructs were transformed into *E. coli* BL21 Ros2pLysS cells (QB3, Macrolab, Berkeley, CA). Dense overnight cultures were grown in 100 µg/mL ampicillin (RPI), 20 µg/mL chloramphenicol (RPI) in LB, and then 12 mL were added to 1 L of TB that contained 100 µg/mL ampicillin and 20 µg/mL chloramphenicol. At A600 of ~0.6–0.8, IPTG was added to a final concentration of 100 µM, and cells were grown overnight at 18°C. Each pellet was then resuspended in 1:4 weight/volume ratio of 50 mM Tris, 30% sucrose, pH 8.0, 1 mM AEBSF, 1 mM benzamidine, 0.1 mg/mL DNAase, and 0.1 mg/mL lysozyme. This suspension was nutated for 30 min at 4°C and then centrifuged at $5000 \times g$ for 55 min. The supernatant was decanted, and then the pellet was resuspended with 1:4 g/mL of ice-cold 5 mM Tris 5 mM MgCl$_2$ pH 8.0. This suspension was nutated for 30 min at 4°C and then centrifuged at $8000 \times g$ for 10 min. The sucrose and Mg supernatants were decanted and further clarified if needed by centrifuging at $4000 \times g$ for 30 min and then combined. The combined supernatant was then run over a gravity column of His60 (Takara, Kusatsu, Shiga, Japan) resin. The column was washed with 20 column volumes (CV) of Buffer A (10 mM imidazole, 50 mM NaPO$_4$, 300 mM NaCl, pH 8.0), then eluted with 5 CV of a stepwise gradient of 40 mM, 70 mM, 125 mM, and 250 mM imidazole. The buffer with 250 mM imidazole is herein referred to as Buffer B. The pure fractions determined by SDS-PAGE Stain-Free gel (Bio-Rad) were combined and concentrated to ~1 mL and dialyzed using 7 KMWCO Snake Skin dialysis tubing (Thermo Scientific) into 20 mM Tris, 150 mM NaCl, pH 8.4. Thrombin agarose beads (15 µL) (Sigma) were washed 3× with 500 µL of Buffer C (20 mM Tris, 150 mM NaCl, 2.5 mM CaCl$_2$, pH 8.4), where the last wash of the resin was incubated for 30 min to activate the thrombin. The dialyzed protein was added to the beads and was gently rocked overnight at RT. The solution was centrifuged for 1 min at $1000 \times g$ and then washed with 2 × 1 mL of Buffer B. The washes and supernatant were combined and run over a His60 bed. The resin was washed with 10 CV of Buffer A and 10 CV of Buffer B. The flowthrough and the Buffer A fractions were combined, concentrated, and dialyzed into 50 mM NaOAc, 50 mM NaCl, pH 5.5, then into 50 mM NaOAc, 50 mM NaCl, 5 µM CuSO$_4$, pH 5.5, and then into 50 mM MOPS, 50 mM NaCl, pH 7.0. The protein was then stored at 4°C and

showed negligible loss of chitin-degrading activity over several months. A protein gel of each of the haplogroup 1–6 purified PMO domains is provided in *Figure 1—figure supplement 4—source data 1*.

## ICP-MS Cu quantification

Protein was diluted to 20 µM and digested in 2% nitric acid overnight at RT in 1.7 mL tubes (Starstedt, Nümbrecht, Germany). The samples were then centrifuged to remove insoluble material at 20,000 × *g* for 10 min, and the supernatant was transferred to new tubes. Samples were analyzed on a Thermo Fisher iCAP Qc ICP mass spectrometer in kinetic energy discrimination (KED) mode against a standard curve of known copper concentrations (CMS-5, Inorganic Ventures, Christiansburg, VA), with Ga (20 µg/L, Inorganic Ventures) as an internal standard. Each experiment was carried out in technical triplicate. Data are plotted with SEM error bars as calculated by GraphPad Prism 9.2.0. ICP data is provided in *Figure 1—source data 2*, *Figure 4—source data 3*, and *Figure 5—source data 3*.

## Purification of *N. crassa* cell walls

Cell wall material was purified as previously described (*Maddi et al., 2009*) with slight modifications: three 1 L flasks of 200 mL, 1.5% agar, 2% sucrose, 1× Vogel's salts were inoculated with conidia from FGSC2489, JW242, Δ*cwr-1*Δ*81*Δ*cwr-2,* and FGSC2489; *cwr-1*$^{\text{D111 HG5}}$ strains and allowed to grow at RT for 7 days with shaking (200 rpm). The conidia were separated from the mycelia with ~200 mL of sterile Milli-Q water and filtered through four layers of cheesecloth. The solution was then centrifuged at 1200 × *g* for 5 min, washed by resuspending the conidia in 15 mL of PBS, and then centrifuged at 1200 × *g* for 5 min. The conidia were then resuspended in PBS to a $1.0 \times 10^8$ conidia/mL titer. The conidia were lysed in a bead beater (BioSpec, Bartlesville, OK) with 0.1 mm glass beads for 5 × 1 min while resting on ice for 2 min in between cycles. Then the mixture was filtered using cheesecloth to remove the beads, and the disrupted conidia were resuspended in 15 mL of PBS. The solution was then centrifuged at 4000 × *g* for 5 min, decanted, and resuspended in 15 mL of PBS with 10% SDS and centrifuged at 4000 × *g* for 5 min. The supernatant was decanted and conidia resuspended in fresh 15 mL of PBS with 10% SDS. The suspension was boiled for 15 min and cooled at RT. The solution was then centrifuged at 10,000 × *g* for 5 min, washed two times with 15 mL of PBS, and then two times with 15 mL Milli-Q water. After gently pipetting away excess water, the disrupted conidia were frozen in liquid nitrogen and lyophilized overnight on a Labconco Freezone 2.5 plus Lyophilizer (Labconco, Kansas City, MO). The dried conidial cell wall material was brought to RT before use.

## Measurement of oxidized products

Chitin (80 mg; α-chitin from powdered shrimp shells; Sigma) and β-chitin (Mahtani Chitosan Pvt. Ltd., Veraval, India) were dispersed and sonicated in 100 mM EDTA for 30 min and washed with Milli-Q water by first resuspending with 1 mL of water and then centrifuging at 1000 × *g* for 10 min. The supernatant was then discarded. The wash procedure was performed five times; then, the resulting chitin was resuspended to a stock concentration of 80 mg/mL. Other substrates examined were tested at 20 mg/mL including PASC that was prepared as previously described (*Wood, 1988*). Purified cell walls were resuspended in 50 mM MOPS, pH 7.0. Then, 10 µM PMO was incubated with 20 mg/mL of the substrate in 50 mM MOPS, pH 7.0. The reactions were initiated with 1 mM ascorbate and incubated for 1 hr, shaking at 37°C. The samples were centrifuged at 20,000 × *g* at 4°C for 10 min, and the supernatant was transferred into new tubes to stop the reaction by removing from substrate. The products were analyzed with HPAEC-PAD using a Dionex ICS-5000 system (Thermo Fisher) with a gradient of three buffers: (A) 10 mM NaOH, (B) 100 mM NaOH, and (C) 100 mM NaOH 500 mM NaOAc. Buffers were prepared as dictated by the Thermo Fisher manual, and the linear gradient (5 curve) was as follows: 0.4 mL/min flow rate: 15–50% B for 15 min; 50–40% B, 5–60% C to 40% B, 60% C for 20 min; isocratic 100% C for 5 min; then 15% B for 15 min. A gold electrode in carbohydrate-quadrupole mode was used as the detector. The autosampler was kept at 4°C to preserve sample integrity. Each protein and substrate combination were analyzed at least in biological triplicate. The HPAEC-PAD source data are provided in *Figure 1—source data 1*, *Figure 4—source data 1*, *Figure 4—source data 2*, *Figure 4—source data 5*, *Figure 4—source data 6*, *Figure 4—source data 7*, *Figure 5—source data 1*, *Figure 5—source data 2*, and *Figure 4—figure supplement 1—source data 1*.

## Generation of C1-oxidized chitooligosaccharide standards

The standards were generated as previously described (*Loose et al., 2014*) with slight modifications. 3 mM of chitooligosaccharides (DP 3–6) (Megazyme, Wicklow, Ireland) were treated with 0.12 mg/mL of the AA3 from *F. graminearum*, ChitO (Gecco Biotech, Groningen, The Netherlands) in 50 mM MOPS, pH 7.0, overnight at RT and then aliquoted and stored at –20°C until use. Source data are provided in *Figure 5—figure supplement 1—source data 1* and *Figure 1—figure supplement 5—source data 2*.

## Liquid chromatography-tandem mass spectrometry

Samples of oligosaccharides were analyzed using a 1200 series liquid chromatography (LC) system (Agilent Technologies, Santa Clara, CA) that was connected in line with an LTQ-Orbitrap-XL mass spectrometer equipped with an electrospray ionization (ESI) source (Thermo Fisher Scientific). The LC system was equipped with a reversed-phase analytical column (length: 150 mm; inner diameter: 1.0 mm; particle size: 5 µm; Viva C18, Restek, Bellefonte, PA). Acetonitrile, formic acid (Optima LC-MS grade, 99.5+%, Fisher, Pittsburgh, PA), and water purified to a resistivity of 18.2 MΩ·cm (at 25°C) using a Milli-Q Gradient ultrapure water purification system (Millipore, Billerica, MA) were used to prepare LC mobile phase solvents. Solvent A was 99.9% water/0.1% formic acid, and solvent B was 99.9% acetonitrile/0.1% formic acid (volume/volume). The elution program consisted of isocratic flow at 1% B for 5 min, a linear gradient to 95% B over 1 min, isocratic flow at 95% B for 4 min, a linear gradient to 1% B over 0.5 min, and isocratic flow at 1% B for 19.5 min, at a flow rate of 150 µL/min. The column compartment was maintained at 25°C, and the sample injection volume was 10 µL. Full-scan, high-resolution mass spectra were acquired in the positive ion mode over the range of mass-to-charge ratio (*m/z*) = 150–2000 using the Orbitrap mass analyzer, in profile format, with a mass resolution setting of 100,000 (at *m/z* = 400, measured at full width at half-maximum peak height [FWHM]). For tandem mass spectrometry (MS/MS or MS$^2$) analysis, selected precursor ions were fragmented using collision-induced dissociation (CID) under the following conditions: MS/MS spectra acquired using the Orbitrap mass analyzer, in centroid format, with a mass resolution setting of 15,000 (at *m/z* = 400, FWHM), normalized collision energy 35%, activation time 30 ms, and activation Q 0.25. Mass spectrometry data acquisition and analysis were performed using Xcalibur software (version 2.0.7, Thermo Fisher Scientific). Source data are provided in *Figure 1—figure supplement 6—source data 1*.

## Electrospray ionization time of flight liquid chromatography mass spectrometry (ESI-TOF LC-MS)

Acetonitrile (Optima grade, 99.9%, Fisher), formic acid (1 mL ampules, 99+%, Pierce, Rockford, IL), and water purified to a resistivity of 18.2 MΩ·cm (at 25°C) using a Milli-Q Gradient ultrapure water purification system (Millipore) were used to prepare mobile phase solvents S3 for LC-MS. ESI-MS of proteins was performed using an Agilent 1260 series liquid chromatograph outfitted with an Agilent 6224 time-of-flight (TOF) LC-MS system (Santa Clara, CA). The LC was equipped with a Proswift RP-4H (monolithic phenyl, 1.0 mm × 50 mm, Dionex) analytical column. Solvent A was 99.9% water/0.1% formic acid (v/v) and solvent B was 99.9% acetonitrile/0.1% formic acid (v/v). PMOs were buffer exchanged into 25 mM ammonium bicarbonate buffer pH 7.5 using Biospin 6 (Bio-Rad) columns according to the manufacturer's protocol and then spinning through a 0.22 µm cellulose acetate centrifugal spin filter. Samples of 1–5 µL were injected onto the column, corresponding to 10 pmol of protein. Following sample injection, a 5–100% B elution gradient was run at a 0.30 mL/min flow rate over 8 min. Data were collected and analyzed by deconvolution of the entire elution profile (using Agilent Mass Hunter Qualitative Analysis B.05.00) to provide reconstructed mass spectra. Spectra were analyzed with open-source Chartograph software (http://chartograph.com/; accessed 3/21/21) and compared to the mass predicted by Benchling (http://chartograph.com/; accessed 3/21/21). Source data are provided in *Figure 1—figure supplement 4—source data 2* and *Figure 5—figure supplement 1—source data 1*.

## O$_2$ reduction measurements

The formation of H$_2$O$_2$ from PMO activity in the absence of polysaccharide substrate was measured using a horseradish peroxidase (HRP)-coupled assay that oxidized Amplex red to produce a colorimetric readout. A PMO sample (2 µM) was incubated at RT with 100 µM Amplex red 1.3 µM HRP

(Sigma), 2 mM ascorbate in 50 mM MOPS, pH 7.0. The absorbance at 540 nm was measured at 30 s intervals for 10 min on SpectraMax340 spectrophotometer (Molecular Devices, San Jose, CA). Each reaction was done in at least biological triplicate. Data are plotted with SEM error bars as calculated by GraphPad Prism 9.2.0. Data for HRP assays are provided in *Figure 1—source data 3*, *Figure 4—source data 4*, and *Figure 5—source data 4*.

## Homology modeling

The sequence of each haplogroup was used without the signal peptide as predicted by SignalP 5.0 (http://www.cbs.dtu.dk/services/SignalP/; accessed 4/24/2021) and modeled in Swiss Prot using the copper-bound structure of *Ao*AA11 (PDB 4MAI) as the template. The resulting models were overlaid and visualized in ChimeraX 1.1.

## SSN generation and MSA analysis

The protein sequence of *cwr-1*$^{HG1}$ (NCU01380) was retrieved from the UniProtKB database and used as a search sequence in the EFI-EST website (*Gerlt et al., 2015*; accessed 4/25/2021) using the blast option with a maximum of 10,000 hits and an e value of 5. These retrieved sequences (~1700) were used to generate an SSN. An alignment score of 63 was used to generate and visualize the SSN in Cytoscape. The SSN data set is available in *Supplementary file 1g*. Source raw data for the SSN are available in *Figure 1—figure supplement 2—source data 1*. ClustalW (accessed 4/25/2021) was used to generate an MSA of the *cwr-1* haplogroup sequences and determine intra-haplogroup sequence ID%.

## EPR measurements

EPR spectra were acquired of 225 µM of CWR-1$^{HG1}$ and 540 µM of CWR-1$^{H78A}$ in 50 mM MOPS, 10% glycerol, pH 7.0 at 40 K. X-band continuous-wave (CW) EPR spectra were recorded on a Bruker ELEXSYS E500 spectrometer equipped with a cylindrical TE011-mode resonator (SHQE-W), an ESR 900-liquid helium cryostat, and an ITC-5 temperature controller (Oxford Instruments, Abingdon, UK). Spectra were acquired under slow-passage nonsaturating conditions.

# Acknowledgements

We thank the Glass laboratory and Marletta laboratory members for critical reading of the manuscript. We acknowledge Daniel Brauer for help in analyzing the whole-protein mass spectra. We kindly thank the C Chang laboratory for the use of their ICP-MS. We want to acknowledge the QB3 mass spectrometry facility and Anthony T Iavarone for his help in analyzing high-resolution mass spectra of the reaction products. The QB3/Chemistry Mass Spectrometry Facility received support from the National Institutes of Health (grant 1S10OD020062-01). AMR-R, NLG, and partially TCD were supported by a National Science Foundation grant (MCB-1818283). TCD was also partially supported by the National Science Foundation grant CHE-1904540.

# Additional information

## Funding

| Funder | Grant reference number | Author |
| --- | --- | --- |
| National Science Foundation | MCB-1818283 | Adriana M Rico-Ramírez N Louise Glass Tyler C Detomasi |
| National Science Foundation | CHE-1904540 | Tyler C Detomasi Michael A Marletta |

The funders had no role in study design, data collection and interpretation, or the decision to submit the work for publication.

## Author contributions

Tyler C Detomasi, Adriana M Rico-Ramírez, Conceptualization, Data curation, Formal analysis, Validation, Investigation, Visualization, Methodology, Writing – original draft, Writing – review and editing;

Richard I Sayler, Data curation, Formal analysis, Validation, Visualization, Methodology; A Pedro Gonçalves, Resources, Formal analysis, Investigation, Methodology, Writing – review and editing; Michael A Marletta, N Louise Glass, Conceptualization, Supervision, Funding acquisition, Methodology, Project administration, Writing – review and editing

**Author ORCIDs**
Tyler C Detomasi http://orcid.org/0000-0003-4390-108X
Adriana M Rico-Ramírez http://orcid.org/0000-0002-4196-8427
Richard I Sayler http://orcid.org/0000-0001-7252-707X
A Pedro Gonçalves http://orcid.org/0000-0003-3133-0013
Michael A Marletta http://orcid.org/0000-0001-8715-4253
N Louise Glass http://orcid.org/0000-0002-4844-2890

**Decision letter and Author response**
Decision letter https://doi.org/10.7554/eLife.80459.sa1
Author response https://doi.org/10.7554/eLife.80459.sa2

## Additional files

### Supplementary files
• MDAR checklist

• Supplementary file 1. Description of plasmids, strains and materials used in this study. (a) Plasmids to transform *N. crassa*. (b) Plasmids to transform *E. coli*. (c) Primers used in this study. (d) *N. crassa* strains used in this work. (e) Intra-haplogroup sequence ID. (f) Substrates used to test CWR-1 activity. (g) Sequence similarity network (SSN) data set.

### Data availability
Materials Availability All strains and plasmids listed in Supplementary file 1a, b and d are available upon request or from the Fungal Genetics Stock Center (https://www.fgsc.net). Primers used in this study are listed in Supplementary file 1c. P value data for Figures 1, 3, 5, 6 and Figure 1—figure supplement 3B are provided in the Figure 1—source data 4, Figure 3—source data 1, Figure 5—source data 5, Figure 6—source data 1, Figure 6—source data 2 and Figure 1—figure supplement 3—source data 1. Data for biochemical analyses of CWR-1 are provided in Figure 1—source data 3; Figure 4—source data 4; Figure 5—source data 4 (HRP oxygen reduction assays); Figure 1—source data 2; Figure 4—source data 3; Figure 5—source data 3 (ICP); Figure 1—source data 1; Figure 4—source data 1; Figure 4—source data 2; Figure 4—source data 5; Figure 4—source data 6; Figure 4—source data 7; Figure 5—source data 1, Figure 5—source data 2, Figure 1—figure supplement 5—source data 1; Figure 1—figure supplement 5—source data 2; Figure 4—figure supplement 1—source data 1 (HPAEC-PAD traces). Data for construction of the SSN is provided in Supplementary file 1g and the raw data as Figure 1—figure supplement 2—source data 1. Original data for Figure 1—figure supplement 4A are provided as Figure 1—figure supplement 4—source data 1. Whole protein MS data are provided in Figure 1—figure supplement 4—source data 2, Figure 5—figure supplement 1—source data 1. EPR source data are provided in Figure 5—figure supplement 2—source data 1. Tandem MS data are provided in Figure 1—figure supplement 6—source data 1.

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
