## [Editor Report]

This fundamental study identifies an important role for a lytic polysaccharide monooxygenase in allorecognition in the filamentous fungus *Neurospora crassa*, which is independent of the catalytic activity of this remarkable class of proteins. The study's findings are compelling, combining microscopy with genetics and biochemistry. The study will be of great interest to fungal biologists and microbiologists, as well as biochemists studying carbohydrate-active enzymes.

---

## [Decision Letter]

**Decision letter after peer review:**

Thank you for submitting your article "A moonlighting function of a chitin polysaccharide monooxygenase, CWR-1, in allorecognition in *Neurospora crassa*" for consideration by *eLife*. Your article has been reviewed by 3 peer reviewers, and the evaluation has been overseen by a Reviewing Editor and Naama Barkai as the Senior Editor. The following individuals involved in the review of your submission have agreed to reveal their identity: Jean-Guy Berrin (Reviewer #1); Katja Salomon Johansen (Reviewer #2).

Essential revisions:

There was a strong consensus among the reviewers that this is a very rigorous and well-executed study. There is a series of suggestions that I ask you to consider as you revise your manuscript.

*Reviewer #1 (Recommendations for the authors):*

I have only a list of comments that may help the authors to improve the manuscript before publication:

– In the abstract please define cwr.

– The term moonlighting has already been used by others in the PMO/LPMO field. In the abstract and line 923, it feels as if the authors were using this term for the first time in this field.

– The authors should mention that PMOs are also called LPMOs.

– Line 29: which domain? The catalytic domain?

– Line 36: CWR-1.

– Several times a space is missing before or after (refs).

– Line 97: Is it the right ref??? Levasseur et al.

– Line 110: Moreau et al. refer to the production of nanocellulose and not to biofuels. Please correct the sentence accordingly.

– Lines 111-115: the authors could quote the review "On the expansion of biological functions of lytic polysaccharide monooxygenases by Vandhana et al. New Phytologist.

– Line 122: What is a domain similar in sequence to X278? Please clarify this statement. Later in the manuscript, the authors make the assumption that the X278 binds to chitin but it would have been good to test the binding of this domain (or full-length PMO) to chitin or fungal cell wall to strengthen the hypothesis.

– Line 123: GH18 (without hyphen).

– Line 345: what is ChitO? Which type of enzyme? A chitooligosaccharide oxidase? From which organism? Which CAZy family?

– Figure 1: The units of the oxygen reduction assay look a bit odd: 540 nm min-1? Same comment in the other figures.

– Line 540-543: a reference is needed (AA9 nomenclature of loops).

– Figure 3B: Are the GS linkers always the same length?

– Figures 5A and B: inactive mutants seem to release some non-oxidized chitin oligomers. Is it due to binding?

– It would be better to name chitooligosaccharides as chitobiose, triose, … and not as trimer, tetramer… which might be misleading. Another way is to mention their degree of polymerisation (DP).

– Line 871: not clear.

– Lines 932-941: I don't understand why the authors speculate that CWR-2 could act as electron donors (like CDH) if the activity of CWR-1 is not required for cell fusion blockage. By the way, *N. crassa* displays a CDH. Do the authors know if the CDH gene is co-regulated with cwr-1?

*Reviewer #2 (Recommendations for the authors):*

Main comment:

The manuscript is quite long. However, because the enzyme activity – or rather lack of- is central to this work, the authors should pay more attention to the recent literature concerning the (lack of) reactivity of similar mono-nuclear proteins. In particular, the work directly comparing the reactivity of LPMO with the X325 (named Bim1) from Cryptococcus and the bacterial copper chaperone CopC is relevant. Importantly, Bim1 functions together with the copper transporter CTRL1. Could the interaction between cwr-1 and cwr-2 turn out to be similar in the sense that it has more to do with copper than with catalysis?

In my view, no further data are required for publication. However, a closer look at the copper affinity and potential redox activity of the mutant cwr-1 seems to be an important follow-up to this study. The "oxygen reduction assay" using amplex red use here is not the best choice.

*Reviewer #3 (Recommendations for the authors):*

I would suggest the authors to better characterise the X278 domain and show that it binds to chitin and/or cell walls of Neurospora. While this is expected, it is interesting to include it to further strengthen the model.

Furthermore, is CWR-1 demonstrated to sit on the outer cell surface? Is there experimental data available to warrant this claim?

Finally, in the model both germlines have a set of proteins involved in cell fusion (as depicted in Figure 7B). Are both systems active and required to get fusion, or is one enough (as shown in 7C)?

---

## [Author Response]

Essential revisions:Reviewer #1 (Recommendations for the authors):I have only a list of comments that may help the authors to improve the manuscript before publication:– In the abstract please define cwr.

Definition added (line 21-22 in tracked changes manuscript).

– The term moonlighting has already been used by others in the PMO/LPMO field. In the abstract and line 923, it feels as if the authors were using this term for the first time in this field.

We did not mean to imply that we were the first to recognize alternate functions of PMOs. We have references to reviews on this topic in the introduction (Vandhana et al. 2022, Hangasky et al. 2020) (Line 111-112) and to clarify that others have used this term to describe the observation that PMOs and related proteins that are not PMOs perform other unrelated functions (line 947-959). Additional text about X325 proteins has also been added to the introduction (line 115-118). Below, in response to reviewer #2, we discuss the definition of “moonlighting” and those comments are equally relevant here. It is worthy of note that true “moonlighting” as defined, namely a single polypeptide with two distinct functions applies to CWR-1, but not to all PMOs whose physiological function is distinct from the catalytic activity and where catalytic activity has yet to be demonstrated.

– The authors should mention that PMOs are also called LPMOs.

We mention this additional nomenclature in line 99, so no change is necessary. Although almost alone in this, we do not use the LPMO nomenclature. The reasoning is simple. After hydroxylation, the unstable glycosyl linkage breaks without enzymatic intervention. There is no evidence for such intervention (for example general base catalysis). An analogous situation is with the cytochrome P450s. For example, when these enzymes hydroxylate a position α to an amine (a dealkylation reaction), the carbinolamine generated falls apart on its own. The enzyme does not participate; hence they are not called lytic cytochrome P450s.

– Line 29: which domain? The catalytic domain?

The PMO domain is the catalytic domain, reworded to “catalytic (PMO) domain” (line 28).

– Line 36: CWR-1.

Corrected.

– Several times a space is missing before or after (refs).

Fixed throughout.

– Line 97: Is it the right ref??? Levasseur et al.

Changed to Gonçalves et al., 2019 (line 96).

– Line 110: Moreau et al. refer to the production of nanocellulose and not to biofuels. Please correct the sentence accordingly.

Reworded to “industrial biofuel and nanocellulose applications” (line 109-110).

– Lines 111-115: the authors could quote the review "On the expansion of biological functions of lytic polysaccharide monooxygenases by Vandhana et al. New Phytologist.

Two reviews have been cited (line 111-112) Vandhana et al., and Hangasky et al.,

– Line 122: What is a domain similar in sequence to X278? Please clarify this statement. Later in the manuscript, the authors make the assumption that the X278 binds to chitin but it would have been good to test the binding of this domain (or full-length PMO) to chitin or fungal cell wall to strengthen the hypothesis.

We agree that chitin binding experiments would strengthen this hypothesis, however, since this domain does not appear to be important for the allorecognition function, these experiments are, in our view, not essential to this paper. We clarified this section (lines 127-130).

– Line 123: GH18 (without hyphen).

Corrected.

– Line 345: what is ChitO? Which type of enzyme? A chitooligosaccharide oxidase? From which organism? Which CAZy family?

This is a commercially obtained enzyme from Fusarium *graminearum* as described in the methods. It belongs to the AA3 CAZy family. Updated lines 358 and 553-54 to make this clear to the reader.

– Figure 1: The units of the oxygen reduction assay look a bit odd: 540 nm min-1? Same comment in the other figures.

Should Be “A _540nm_” as in figure 5D. Figures 1 and 4 were edited with the appropriate units for the y-axis.

– Line 540-543: a reference is needed (AA9 nomenclature of loops).

We thank the reviewer for noticing this oversight; we added Danneels et al., 2018 and Liu B. et al., 2008 (line 573).

– Figure 3B: Are the GS linkers always the same length?

The GS linkers are ~100 amino acids long between the CWR-1 proteins from the different haplogroups, with a range from 80 to 116 amino acids. In this figure, the length of the GS linker in the chimeric constructions is identical, as we used the glycine serine linker domain from the HG1 strain (FGSC2489).

– Figures 5A and B: inactive mutants seem to release some non-oxidized chitin oligomers. Is it due to binding?

We appreciate the fine attention to detail. Some of the signal is due to ascorbate. There may be very minor amounts of non-oxidized oligomers due to weak hydrolytic activity as observed with other PMOs or as the reviewer suggests may be from binding to the substrate. However, the amount present does not appear to be relevant when compared to WT traces and, most importantly, there are no oxidized peaks present.

– It would be better to name chitooligosaccharides as chitobiose, triose, … and not as trimer, tetramer… which might be misleading. Another way is to mention their degree of polymerisation (DP).

We agree and to avoid confusion have updated the suffixes to -ose to better specify the sugar versus other potential species in the Figure Legend to Figure 1—figure supplement 5. Oligosaccharide standard chromatograms and elution times.

– Line 871: not clear.

We included edits throughout the manuscript to clarify the comparison of PMO domains from single domain to multi domain architectures (line 894-896).

– Lines 932-941: I don't understand why the authors speculate that CWR-2 could act as electron donors (like CDH) if the activity of CWR-1 is not required for cell fusion blockage. By the way, *N. crassa* displays a CDH. Do the authors know if the CDH gene is co-regulated with cwr-1?

After consideration of the reviewers’ comments, we agree that this speculation is not related or necessary for the story and could cause confusion. We therefore removed this entire paragraph from the discussion. CDHs have been shown to be partners with AA9s and are active on cellobiose, which matches the cellulose substrate of AA9 enzymes. In the context of AA11 enzymes, it is not clear why CDH would be present with AA11s that are active on chitin substrates. Additional experimentation is needed to make a more definitive claim.

Reviewer #2 (Recommendations for the authors):Main comment:The manuscript is quite long. However, because the enzyme activity – or rather lack of- is central to this work, the authors should pay more attention to the recent literature concerning the (lack of) reactivity of similar mono-nuclear proteins. In particular, the work directly comparing the reactivity of LPMO with the X325 (named Bim1) from Cryptococcus and the bacterial copper chaperone CopC is relevant. Importantly, Bim1 functions together with the copper transporter CTRL1. Could the interaction between cwr-1 and cwr-2 turn out to be similar in the sense that it has more to do with copper than with catalysis?

The reviewer brings up a good point. Bim1 (and other X325 proteins), while sharing a similar fold and copper binding site, are not a PMO, and to date catalytic activity has not been observed on any polysaccharide substrate. CWR-1 is active on chitin, and has an additional role in allorecognition, which does not involve catalysis. CWR-1 is not involved in copper transport since a variant that no longer binds copper is still functional in allorecognition, whereas in Bim1, the copper site is required for transport. We have referenced this class of proteins that are not PMOs in the introduction (lines 115-118).

In my view, no further data are required for publication. However, a closer look at the copper affinity and potential redox activity of the mutant cwr-1 seems to be an important follow-up to this study. The "oxygen reduction assay" using amplex red use here is not the best choice.

Thank you and we agree with this comment. The Amplex Red assay shows clearly that the enzyme is competent for oxygen reduction. The double His mutant is the essential variant for this paper and no longer binds copper. It would, of course, have no oxygen reduction activity.

Reviewer #3 (Recommendations for the authors):I would suggest the authors to better characterise the X278 domain and show that it binds to chitin and/or cell walls of Neurospora. While this is expected, it is interesting to include it to further strengthen the model.

We agree that this is an interesting and worthwhile experiment, however since the X278 domain does not appear to be important for the observed phenotype this will be pursued in follow up studies as is not directly relevant to the process of allorecognition.

Furthermore, is CWR-1 demonstrated to sit on the outer cell surface? Is there experimental data available to warrant this claim?

CWR-1 has a signal peptide and predicted cleavage at the histidine site that is common in PMO that are secreted. We were unable to observe GFP-labeled CWR-1 in live cells using the native promoter, but observed both intracellular and membrane localization in an overexpression strain. To fully understand the localization of both CWR-1 and CWR-2 during growth and during a fusion block, we believe that immunogold labeling and TEM are necessary, which will be pursued in further studies.

Finally, in the model both germlines have a set of proteins involved in cell fusion (as depicted in Figure 7B). Are both systems active and required to get fusion, or is one enough (as shown in 7C)?

In wild type cells, both CWR-1 and CWR-2 are present, making it possible that the fusion block could be triggered by two incompatible CWR-1/CWR-2 interactions. In our simplified system, we showed that, in otherwise isogenic strains, one incompatible CWR-1/CWR-2 interaction is sufficient to trigger allorecognition and a block in cell fusion and which function *in trans* (CWR-1 in one cell in interactions with incompatible CWR-2 in a separate cell is sufficient to trigger cell fusion arrest). While both incompatible combinations of CWR-1 and CWR-2 are present in wild type cells, Figure 7C simplifies this model to explain what was shown in this study. One CWR-1/CWR-2 mismatch is enough, but we note in the text and figure legend that both are present and result in a potential redundancy in this mismatch recognition.